# PieClam: A Universal Graph Autoencoder
# Based on Overlapping Inclusive and Exclusive Communities

**Daniel Zilberg** [1]   **Ron Levie** [1]

## Abstract

We propose PieClam (Prior Inclusive Exclusive Cluster Affiliation Model): a graph autoencoder, where nodes are embedded into a code space by an algorithm that maximizes the log-likelihood of the decoded graph. PieClam is a community affiliation model that extends well-known methods like BigClam in two main manners. First, instead of the decoder being defined via pairwise interactions between the nodes in the code space, we also incorporate a learned prior on the distribution of nodes in the code space, turning our method into a graph generative model. Secondly, we generalize the notion of communities by allowing not only sets of nodes with strong connectivity, which we call inclusive communities, but also sets of nodes with strong disconnection, which we call exclusive communities. By introducing a new graph similarity measure, called the log cut distance, we show that PieClam is a universal autoencoder, able to uniformly approximately reconstruct any graph. Our method is shown to obtain competitive performance in graph anomaly detection and link prediction benchmarks.

## 1. Introduction

In recent years, considerable research has concentrated on graph representation learning, aiming to develop vector representations for graph entities, including nodes, edges, and subgraphs (Hamilton et al., 2017; Chen et al., 2020a). In graph autoencoders, e.g., (Kipf & Welling, 2016; Grover et al., 2019; Samanta et al., 2020; Mehta et al., 2019), the vertices of a graph are embedded in a code space, where edges are inferred from the locations of the vertices in this space. Encoding graphs into a standard space has a number

of advantages. While different graphs can have different sizes and topologies, the code space is fixed with a fixed dimension. This helps when learning downstream tasks, where the representation of the graph in the code space can be processed. For example, in link prediction, one infers unknown edges by defining (Kipf & Welling, 2016) or learning (Kumar et al., 2020) a function that takes pairs of nodes in the code space and predicts if there is an edge between them. In anomaly detection, one defines (Ding et al., 2019a; Fan et al., 2020) or learns (Chen et al., 2020b; Ding et al., 2021) a function that takes the representation of a node and its neighborhood and predicts if this node is normal or an anomaly. In graph and node classification, the graph is represented in the code space, and one learns a model that predicts from this representation the classes of the nodes or of the graph (Kipf & Welling, 2017; Gilmer et al., 2017a).

**Our Contribution.**  In this paper, we derive a new graph autoencoder from a statistical model of graphs. In our model, similarly to Stochastic Block Models (SBM) (Nowicki & Snijders, 2001b; Lee & Wilkinson, 2019) and community-based statistical models (Airoldi et al., 2008; Yang & Leskovec, 2013), graphs are generated from a combination of intersecting communities/cliques. Namely, each node belongs to a different subset of a predefined set of communities, and the affiliations of the nodes to the different communities determine the probabilities of the edges of the graph. Here, the estimation of the community affiliations is seen as an encoder of graphs to a *community affiliation space*, and the computation of the corresponding edge probabilities is seen as a decoder. As opposed to past works, we consider two types of *generalized communities*. First, standard *inclusive communities*, where any two nodes in the same community are likely to be connected. Second, we propose *exclusive communities*, where belonging to the same community reduces the probability of nodes being connected.

For illustration, consider a social network of employers/recruiters and employees/job-seekers. Such a network has roughly bipartite components, where job-seekers do not tend to connect to other seekers, and employers do not connect to other employers, but employers and seekers of the same subsector tend to connect. Hence, inclusive com-

---

[1]Faculty of Mathematics, Technion - Israel Institute of Technology, Haifa, Israel. Correspondence to: Daniel Zilberg <dannyzilberg@gmail.com>, Ron Levie <levieron@technion.ac.il>.

*Proceedings of the 42nd International Conference on Machine Learning*, Vancouver, Canada. PMLR 267, 2025. Copyright 2025 by the author(s).

munities for such a graph can correspond to job titles or subsectors, and exclusive communities can correspond to sets of job-seekers and sets of employees from the same sector. For example, while the inclusive community of *product management* will tend to connect all employers and specialists of product management to a "clique-like" community, the two exclusive communities of employers and of specialists of product management will each tend to delete the links between pairs of employers and between pairs of employees, creating a bipartite structure.

To formalize the above ideas, we propose the *Prior Inclusive Exclusive Cluster Affiliation Model (PieClam)*. This model represents graphs as overlapping inclusive and exclusive communities. In addition to embedding the nodes of a given graph in the community code space, our model also learns a prior on the code space, so new graphs can be generated from the learned model. This makes PieClam a graph generative model, like, e.g., (Kipf & Welling, 2016; Grover et al., 2019; Samanta et al., 2020; Mehta et al., 2019; Sun et al., 2019).

To model both types of communities, we propose a new type of decoder based on the *Lorentz inner product*, in which case the code space is typically called a *pseudo Euclidean space* (Greub, 1963). The addition of exclusive communities in our model is not merely aimed at improving it heuristically for special graphs like social networks. Rather, we prove in Theorems 3.7 and 3.8 that using exclusive communities (via the Lorentz inner product) makes our model universal, namely, able to approximate with a fixed budget of parameters any graph. This is in contrast to standard decoders based on merely inclusive communities, which we show are unable to represent many graphs.

To formalize this universality property, we propose a new similarity measure between graphs with edge probabilities, that we call the *log cut distance*. We formalize the universality of our model as follows: one can choose the dimension of the code space (the number of communities) a priori, and guarantee that any graph of any size and topology can be approximated up to small log cut distance by decoding points from this fixed space. We show that other related decoders do not satisfy this universality property.

In Section 4 we support our construction with experiments, where our models achieve competitive performance with respect to state of the art. First, we use PieClam to perform graph anomaly detection. Here, PieClam learns a probabilistic model given graph, and this model can be used for inspecting the probabilities of different nodes in this graph: nodes with low probabilities are deemed to be anomalies. Then, we use PieClam to predict edges in link prediction benchmarks.

Appendix B has an extended discussion on related work.

## 2. Community Affiliation Models With Prior

### 2.1. Notations

We denote by $\mathbb{R}_+$ the non-negative real numbers. We denote "and" by $\wedge$. We denote matrices as boldface capital letter $\mathbf{B} = \{b_{n,m}\}_{n,m}$, vectors as boldface lowercase letters $\mathbf{b} = \{b_n\}_n$, and scalars as lowercase letters $b$. Vectors $\mathbf{b} \in \mathbb{R}^N$ are always column vectors. The rows of a matrix $\mathbf{B} \in \mathbb{R}^{N \times C}$ are denoted by the same letter in lowercase $\mathbf{b}_n^\top$, where $\mathbf{b}_n \in \mathbb{R}^C$, where we write in short $\mathbf{b}_n^\top \in \mathbb{R}^C$. A diagonal matrix $\mathbf{D} \in \mathbb{R}^{N \times N}$ with diagonal entries $\mathbf{d}$ is denoted by $\mathrm{diag}(\mathbf{d}) = \mathrm{diag}(d_1, \ldots, d_N)$. The $\ell^2$ norm of a vector $\mathbf{b} \in \mathbb{R}^N$ is defined to be $\|\mathbf{b}\| = (\sum_{n=1}^N b_n^2)^{1/2}$.

A graph is denoted by $G = ([N], E, \mathbf{A})$, where $[N] = \{1, \ldots, N\}$ is the set of $N$ nodes, $E \subseteq [N] \times [N]$ is the set of edges, and $\mathbf{A} \in \{0, 1\}^{N \times N}$ is the adjacency matrix. For weighted graphs, $\mathbf{A} \in [0, 1]^{N \times N}$, where $a_{n,m}$ is the edge weight of $(n, m)$. In this work we focus on undirected graphs, for which $(m, n) \in E \Leftrightarrow (n, m) \in E$, and $\mathbf{A} = \mathbf{A}^\top$. Any pair $(n, m) \in [N] \times [N]$ is called a *dyad*. The neighborhood of a node $n \in [N]$ is $\mathcal{N}(n) = \{m \in [N] \mid (m, n) \in E\}$. A graph-signal is a graph with node features $G = ([N], E, \mathbf{A}, \mathbf{X})$ where $\mathbf{X} = \{\mathbf{x}_n^\top\}_{n=1}^N \in \mathbb{R}^{N \times D}$, $\mathbf{x}_n \in \mathbb{R}^D$, and $D$ is called the feature dimension. A random graph is a graph-valued random variable. Given a random graph with nodes $[N]$, we denote the event $(n, m) \in E$ by $n \sim m$, and the event $(n, m) \notin E$ by $\neg(n \sim m)$. A graph is bipartite if its vertex set $[N]$ can be partitioned into two disjoint sets $\mathcal{U}$ and $\mathcal{V}$, with $\mathcal{U} \cup \mathcal{V} = [N]$, such that every edge has one endpoint in $\mathcal{U}$ and the other in $\mathcal{V}$.

### 2.2. BigClam

Our model PieClam is best understood as an extension of the BigClam model (Yang & Leskovec, 2012). The code space in BigClam is $\mathbb{R}_+^C$, where each axis is interpreted as a community. Each entry $f^c$ of a point $\mathbf{f} = (f^1, \ldots, f^C) \in \mathbb{R}_+^C$ is interpreted as how much the point belongs to community $c$, where 0 means "not included in $c$" and 1 "included." In this paper we call $\mathbb{R}_+^C$ *the affiliation space (AS)*, and call any point in the affiliation space an *affiliation feature (AF)*.

BigClam is a model where a simple random graph is decoded from a sequence of AFs $\mathbf{F} = \{\mathbf{f}_n = (f_n^1, \ldots, f_n^C)\}_{n=1}^N$ in the AS. For each pair of nodes $n \neq m \in [N]$, the probability of the event $\neg(n \sim m)$ (no edge between $n$ and $m$), given their amount of membership in the same community $c$, is defined to be

$$P(\neg(n \sim m)|f_n^c, f_m^c) = e^{-f_n^c f_m^c}.$$

BigClam makes two assumptions of independence. First, the membership in one community does not affect the mem-

bership in another community. Hence,

$$P(\neg(n \sim m))|\mathbf{F}) = P(\neg(n \sim m))|\mathbf{f}_n, \mathbf{f}_m) = e^{-\mathbf{f}_n^\top \mathbf{f}_m},$$
(1)
$$P(n \sim m)|\mathbf{F}) = P(n \sim m|\mathbf{f}_n, \mathbf{f}_m) = 1 - e^{-\mathbf{f}_n^\top \mathbf{f}_m}.$$

Due to this formula, BigClam is a so called *Bernoulli-Poisson model* (see Appendix B.4 for more details). The second assumption is that the events of having an edge between different pairs of nodes are independent. As a result, the probability of the entire graph $([N], E)$ conditioned on the AFs $\mathbf{F}$ is

$$P(E|\mathbf{F}) =$$
$$\prod_{n \in [N]} \left( \sqrt{ \prod_{m \in \mathcal{N}(n)} P(n \sim m|\mathbf{F}) \prod_{m \notin \mathcal{N}(n)} P(\neg(n \sim m)|\mathbf{F}) } \right)$$
(2)

Here, the square root is taken because the product considers each edge twice, and all sums are over $m \neq n$.

BigClam consists of a decoder, decoding from the AFs $\mathbf{F}$ the random graph $G(\mathbf{F})$ with node set $[N]$ and independent Bernoulli distributed edges with probabilities $\mathbf{P} = \{P(n \sim m|\mathbf{f}_n, \mathbf{f}_m)\}_{n,m=1}^N$.

From the encoding side, given a simple graph $([N], E)$, BigClam encodes the graph into the affiliation space by maximizing the log likelihood with respect to the AFs $\mathbf{F}$

$$l(\mathbf{F}) = \frac{1}{2} \sum_{n \in [N]} \left( \sum_{m \in \mathcal{N}(n)} \log(1 - e^{-\mathbf{f}_n^\top \mathbf{f}_m}) - \sum_{m \notin \mathcal{N}(n)} \mathbf{f}_n^\top \mathbf{f}_m \right),$$
(3)

where all sums are over $m \neq n$. BigClam is optimized by gradient descent, with update at iteration $i$

$$\mathbf{F}^{(i+1)} = \mathbf{F}^{(i)} + \delta \nabla_\mathbf{F} l(\mathbf{F}^{(i)}),$$
(4)

for some learning rate $\delta > 0$. In order to implement the above iteration with $O(|E|)$ operations at each step, instead of $O(N^2)$, the loss can be rearranged as

$$2l(\mathbf{F}) =$$
$$\sum_{n \in [N]} \left( \sum_{m \in \mathcal{N}(n)} \log(e^{\mathbf{f}_n^\top \mathbf{f}_n} - 1) - \mathbf{f}_n^\top \sum_{n \in [N]} \mathbf{f}_m + \|\mathbf{f}_n\|^2 \right).$$
(5)

The gradient of the loss is now

$$\nabla_{\mathbf{f}_n} l = \sum_{m \in \mathcal{N}(n)} \mathbf{f}_m \left( 1 - e^{-\mathbf{f}_n^\top \mathbf{f}_m} \right)^{-1} - \sum_{n \in [N]} \mathbf{f}_m + \mathbf{f}_n.$$
(6)

Since the global term needs to be calculated only once, the number of operations is $O(|E|)$ instead of $O(N^2)$.

We observe that the optimization process is a message passing scheme. Looking at the dynamics of the optimization process, we see that every node is pushed in the direction of a weighted average of all of its neighbors, and pushed in the opposite direction to the average of all of the nodes. The sum of both forces tends to drive communities towards the axes in the optimization dynamics.

## 2.3. Inclusive-Exclusive Cluster Affiliation Model

The BigClam decoder has a limitation due to a "triangle inequality type" behavior. Namely, suppose that we would like to construct two features $\mathbf{f}_1$ and $\mathbf{f}_2$ with strong connectivity to a third feature $\mathbf{f}_3$, and we are limited by a fixed affiliation feature dimension $C$. A naive approach to achieve this would be to put $\mathbf{f}_1$ and $\mathbf{f}_2$ close to $\mathbf{f}_3$ so they have large inner products $\mathbf{f}_1^\top \mathbf{f}_3$ and $\mathbf{f}_2^\top \mathbf{f}_3$. This would mean that $\mathbf{f}_1^\top \mathbf{f}_2$ would also be large, so $\mathbf{f}_1$ would be strongly connected to $\mathbf{f}_2$ under the BigClam model. However, some graphs, like bipartite graphs, do not exhibit this triangle inequality type behavior. For a rigorous treatment, see Section 3.4 and Appendix A.1. Next, we build the *IeClam* decoder, that allows decoding any graph, that can model bipartite components without being limited by a triangle inequality-type behavior.

Our *Inclusive Exclusive Cluster Affiliation Model (IeClam)* can be extended from BigClam by replacing the inner product in the non-edge probability (1) by the more expressive *Lorentz inner product*, which does not enforce a triangle inequality-type behavior. For that, we extend the affiliation space (AS) to be $\mathbb{R}^{2C}$, with two types of communities. The first $C$ axes are called the *inclusive communities*, and their corresponding features are called *inclusive affiliation features (IAF)*, denoted by $\mathbf{t} \in \mathbb{R}^C$. The last $C$ axes are called the *exclusive communities*, with *exclusive affiliation features (EAF)*, denoted by $\mathbf{s} \in \mathbb{R}^C$.[1] We define the concatenated affiliation feature by $\mathbf{f} = (\mathbf{t}, \mathbf{s}) \in \mathbb{R}^{2C}$. Given a sequence of affiliation features $\mathbf{F} = \{\mathbf{f}_n\}_{n=1}^N \in \mathbb{R}^{N \times 2C}$, IeClam defines the probability of a single edge by

$$P(n \sim m|\mathbf{f}_n, \mathbf{f}_m) = 1 - \exp\left(-\mathbf{t}_n^\top \mathbf{t}_m + \mathbf{s}_n^\top \mathbf{s}_m\right)$$
$$= 1 - \exp\left(-\mathbf{f}_n^\top \mathbf{L} \mathbf{f}_m\right),$$
(7)

where $\mathbf{L} = \text{diag}(1, ...1, -1, ... - 1)$. The bilinear form $(\mathbf{u}, \mathbf{v}) \mapsto \mathbf{u}^\top \mathbf{L} \mathbf{v}$ is called the *Lorentz inner product*, and has its roots in special relativity. See Appendix B.9 for more details on the Lorentz inner product.

Note that $\mathbf{L}$ is not positive-definite, so it does not actually define an inner product. Moreover, $\mathbf{f}_n^\top \mathbf{L} \mathbf{f}_m$ can be negative even if $\mathbf{f}_n, \mathbf{f}_m \in \mathbb{R}_+^C$. To guarantee that (7) defines a proper probability between 0 and 1, we limit the affiliation space as follows.

**Definition 2.1.** A *cone of non-negativity* is a subset $\mathcal{C}$ of $\mathbb{R}^{2C}$ such that every $\mathbf{f}, \mathbf{g} \in \mathcal{C}$ satisfy $\mathbf{f}^\top \mathbf{L} \mathbf{g} \geq 0$.

If we limit the affiliation space to be a cone of non-negativity, then IeClam gives well defined edge probabilities in $[0, 1]$. In our experiments, we restrict ourselves to the following simple construction of a cone of non-negativity, noting that it is not the only possible construction.

---

[1] More generally, one can define a different number of inclusive and exclusive communities.

**Definition 2.2.** The *pairwise cone* $\mathcal{T}^C$ is defined to be the set of affiliation features $\mathbf{f} = (\mathbf{t}, \mathbf{s}) \in \mathbb{R}^{2C}$ such that for every $c \in [C]$ we have $-t^c \leq s^c \leq t^c$.

It is easy to see that $\mathcal{T}^C$ is a cone of non-negativity. Indeed, for any $\mathbf{f}_1 = (\mathbf{t}_1, \mathbf{s}_1), \mathbf{f}_2 = (\mathbf{t}_2, \mathbf{s}_2) \in \mathcal{T}^C$, we have

$$\forall c \in [C]: \quad t_1^c t_2^c - s_1^c s_2^c \geq 0,$$

so

$$\mathbf{f}_1^\top \mathbf{L} \mathbf{f}_2 = \sum_{c=1}^C t_1^c t_2^c - s_1^c s_2^c \geq 0.$$

As opposed to the BigClam decoder, the IeClam model can approximate a bipartite graph with a small number of communities. Namely, a bipartite graph with two (disjoint) sides $\mathcal{U}, \mathcal{V} \subset [N]$ and constant probability for edges between $\mathcal{U}$ and $\mathcal{V}$ can be represented using $C = 1$. Here, all of the nodes in $\mathcal{U}$ are encoded to $(a, a)$, and all of the nodes of $\mathcal{V}$ are encoded to $(a, -a)$, for some $a \geq 0$. This gives zero probability for edges within each part, and probability $1 - e^{-2a^2}$ for edges between the parts.

*Remark* 2.3. The above construction gives an interpretation for each pair of axes $(t^c, s^c)$ in $\mathcal{T}^C$ as a *generalized community* which can model anything between a clique and a bipartite component.

Given AFs $\mathbf{F}$ in a cone, IeClam is seen as a decoder, by decoding $\mathbf{F}$ into the random graph $G(\mathbf{F})$ with node set $[N]$ and independent Bernoulli distributed edges with probabilities $\mathbf{P} = \{P(n \sim m | \mathbf{f}_n, \mathbf{f}_m)\}_{n,m=1}^N$.

For affiliation features $\mathbf{F} \in \mathcal{T}^{C \times N}$, the probability of the graph $([N], E)$ given $\mathbf{F}$ of IeClam is

$$P(E|\mathbf{F})$$
$$= \prod_{n \in [N]} \left( \sqrt{\prod_{m \in \mathcal{N}(n)} (1 - e^{-\mathbf{f}_n^\top \mathbf{L} \mathbf{f}_m})} \prod_{m \notin \mathcal{N}(n)} e^{-\mathbf{f}_n^\top \mathbf{L} \mathbf{f}_m} \right). \quad (8)$$

Like BigClam, IeClam is optimized by maximizing the log likelihood with gradient descent

$$2l(\mathbf{F}) =$$
$$\sum_{n \in [N]} \left( \sum_{m \in \mathcal{N}(n)} \log(1 - e^{-\mathbf{f}_n^\top \mathbf{L} \mathbf{f}_m}) - \sum_{m \notin \mathcal{N}(n)} \mathbf{f}_n^\top \mathbf{L} \mathbf{f}_m \right) \quad (9)$$

This loss can be efficiently implemented on sparse graphs, with $O(|E|)$ complexity, by the formulation

$$2l(\mathbf{F}) =$$
$$\sum_{n \in [N]} \left( \sum_{m \in \mathcal{N}(n)} \log(e^{\mathbf{f}_n^\top \mathbf{L} \mathbf{f}_m} - 1) - \mathbf{f}_n^\top \mathbf{L} \sum_{m \in [N]} \mathbf{f}_m + \mathbf{f}_n^\top \mathbf{L} \mathbf{f}_n \right). \quad (10)$$

The gradient of the loss for node $n$ is

$$\nabla_{\mathbf{f}_n} l = \mathbf{L} \left( \sum_{m \in \mathcal{N}(n)} \mathbf{f}_m (1 - e^{-\mathbf{f}_n^\top \mathbf{L} \mathbf{f}_m})^{-1} - \sum_{m \in [N]} \mathbf{f}_m + \mathbf{f}_n \right). \quad (11)$$

Notice that all of the calculations are the same as BigClam, up to replacing the dot product by the Lorenz inner product. Hence, the optimization process of IeClam is a message passing scheme, and equivariant to node re-indexing.

## 2.4. Community Affiliation Models With Prior

BigClam and IeClam are not generative graph models. Indeed, these methods only fit a conditional probability of the graph, conditioned on the AF values, but the methods do not learn the probability of the AFs over the affiliation space. Hence, the total probability of $E \wedge \mathbf{F}$ is not defined.

To extend IeClam (and similarly BigClam) into probabilistic generative models, we define a prior probability distribution over the affiliation cone space $\mathcal{C} \subset \mathbb{R}^{2C}$, with probability density function $p : \mathbb{R}^{2C} \to [0, \infty)$ supported on $\mathcal{C}$. Using Bayes law, we now obtain the joint probability $P(E \wedge \mathbf{F})$ of the edges and community affiliation features via the probability density function

$$p(E, \mathbf{F}) = P(E|\mathbf{F})p(\mathbf{F}).$$

We assume that the prior probabilities of all nodes are independent, namely,

$$p(\mathbf{F}) = \prod_{n \in [N]} p(\mathbf{f}_n).$$

Hence, the probability densities that a dyad $(n, m)$ is an edge or non-edge are

$$p(n \sim m, \mathbf{f_n}, \mathbf{f_m}) = p(\mathbf{f}_n) p(\mathbf{f}_m)(1 - e^{-\mathbf{f}_n^\top \mathbf{L} \mathbf{f}_m}),$$

$$p(\neg(n \sim m), \mathbf{f_n}, \mathbf{f_m}) = p(\mathbf{f}_n) p(\mathbf{f}_m) e^{-\mathbf{f}_n^\top \mathbf{L} \mathbf{f}_m}.$$

As before, we assume that the probabilities of different edges are independent, which gives

$$p(E, \mathbf{F}) =$$
$$\prod_{n \in [N]} p(\mathbf{f}_n) \sqrt{\prod_{m \in \mathcal{N}(n)} P(n \sim m | \mathbf{F}_m) \prod_{m \notin \mathcal{N}(n)} P(\neg(n \sim m) | \mathbf{f}_n, \mathbf{f}_m)}. \quad (12)$$

Now, the log likelihood loss is

$$l(\mathbf{F}) = \sum_{n \in [N]} \Bigg( \log(p(\mathbf{f}_n)) +$$
$$\frac{1}{2} \left( \sum_{m \in \mathcal{N}(n)} \log(e^{\mathbf{f}_n^\top \mathbf{L} \mathbf{f}_m} - 1) - \mathbf{f}_n^\top \mathbf{L} \sum_{m \in [N]} \mathbf{f}_m + \mathbf{f}_n^\top \mathbf{L} \mathbf{f}_n \right) \Bigg). \quad (13)$$

We call this extension of IeClam PieClam (Prior Inclusive Exclusive Cluster Affiliation Model). We similarly extend BigClam to PClam (Prior Cluster Affiliation Model) by replacing $\mathbf{L}$ in (13) with the identity matrix.

Observe that the PieClam loss is similar to the IeClam loss, only with the addition of the prior acting as a per node

regularization term. The prior attracts all nodes to areas of higher probability during the optimization dynamics. As before, the optimization process of PieClam is a message passing scheme with $O(|E|)$ complexity, and is equivariant to node re-indexing. Also, note that adding the prior to the Clam model only takes $O(|N|)$ complexity, which is lower than the complexity of the IeClam terms.

In order to sample from the above generative models, we first sample features $\{\mathbf{f}_n\}_{n\in[N]}$ according to $p$, and then connect them using either the BigClam or IeClam conditional probability. To model the prior in practice, we use *realNVP*, which is a *normalizing flow* neural network model (Dinh et al., 2016). For more details on normalizing flows, see Appendix B.8.

**PieClam for graphs with node features.** So far, we have used only the topology of the graph, not considering node features. We extend PieClam (and PClam) to graph-signals $([N], E, \mathbf{X})$ as follows. Concatenate the feature space of $\mathbf{X}$ to the affiliation space, and learn the prior on this combined space. This only affects the prior $p$. The conditional edge probabilities are defined only in terms of the affiliation features, as before.

# 3. Universality of PieClam and IeClam

In this section, we define the universality of graph autoencoders, and prove that IeClam and PieClam are universal, while BigClam and PClam are not. The motivation behind the universality definition is that we would like to uniformly choose the dimension of the code space, such that every graph can be approximated by decoding some points in this fixed code space. Namely, we would like one *universal* decoder that works for all graphs, as opposed to choosing the dimension of the code space depending on the graph.

## 3.1. General Graph Autoencoders

Next, we define a general decoder that defines edge probabilities by operating on pairs of points in a code space.

**Definition 3.1.** A *pairwise decoder* over the code spaces $\mathbb{R}^M$ is a mapping $D_M : \mathbb{R}^{2M} \to [0,1]$. Given $N$ points in the code space $\mathbf{z} = \{z_n \in \mathbb{R}^M\}_{n=1}^N$, the decoded graph $G_N(\mathbf{z})$ is the weighted graph with adjacency matrix

$$\mathbf{D}_M(\mathbf{z}) = \big(D_M(z_n, z_k)\big)_{n,k=1}^N.$$

Clam models are special cases of pairwise decoders.

## 3.2. Log Cut Distance

Our definition of universality has the following form: for every error tolerance $\epsilon > 0$, there is a choice of the dimension $M(\epsilon)$ of the code space such that every graph can be

approximated up to error $\epsilon$ by decoding some points in this space. To formalize the "up to error $\epsilon$" statement, we present in this section a new graph similarity measure which we call the log cut distance. Our construction is based on a well-known graph similarity measure called the cut norm.

**Definition 3.2.** The *cut norm* of a matrix $\mathbf{X} \in \mathbb{R}^{N \times N}$ is defined to be

$$\|\mathbf{X}\|_\square := \frac{1}{N^2} \sup_{\mathcal{U}, \mathcal{V} \subset [N]} \Big| \sum_{i\in\mathcal{U}} \sum_{j\in\mathcal{V}} x_{i,j} \Big|. \qquad (14)$$

The *cut metric* $\|\mathbf{A} - \mathbf{B}\|_\square$ between two adjacency matrices $\mathbf{A}$ and $\mathbf{B}$ is interpreted as the difference between the edge densities of $\mathbf{A}$ and $\mathbf{B}$ on the block $\mathcal{U} \times \mathcal{V}$ on which their edge densities are the most different.

The following graph similarity measure modifies the cut norm, making it appropriate for graphs with random edges over a fixed node set.

**Definition 3.3.** Given two random graphs over the nodes set $[N]$, with independently Bernoulli distributed edges with probabilities $\mathbf{P} = \{p_{n,m}\}_{n,m\in[N]}$ and $\mathbf{Q} = \{q_{n,m}\}_{n,m\in[N]}$ respectively, their *log cut distance* is defined to be

$$D_\square(\mathbf{P}||\mathbf{Q}) := \inf_{0 < e, d \leq 1} \Big( e + d +$$

$$\frac{1}{N^2} \sup_{\mathcal{U}, \mathcal{V} \subset [N]} \Big| \log \Big( \prod_{n\in\mathcal{U}} \prod_{m\in\mathcal{V}} \frac{1 - (1-e)p_{n,m}}{1 - (1-d)q_{n,m}} \Big) \Big| \Big). \qquad (15)$$

The second term in (15) is the cut distance $\|\tilde{\mathbf{P}} - \tilde{\mathbf{Q}}\|_\square$ between the matrix $\tilde{\mathbf{P}}$ with entries

$$\tilde{p}_{n,m} = -\log(1 - (1-e)p_{n,m})$$

and the matrix $\tilde{\mathbf{Q}}$ with entries

$$\tilde{q}_{n,m} = -\log(1 - (1-d)q_{n,m}).$$

Namely, the cut distance between the log likelihoods of non-edges. The parameters $e$ and $d$ make the $[0,1]$-valued probabilities valid inputs to the log. The goal of $e, d$ is to regularize the probability of the edges, where higher regularization is penalized via the additive term $e + d$.

For each choice of a cut $\mathcal{U}, \mathcal{V} \subset [N]$, the term

$$\frac{1}{N^2} \log \Big( \prod_{n\in\mathcal{U}} \prod_{m\in\mathcal{V}} \frac{1 - (1-e)p_{n,m}}{1 - (1-d)q_{n,m}} \Big) \qquad (16)$$

is somewhat similar in structure to an un-normalized KL divergence, or distance of log likelihoods, between the non-edge probabilities of the graphs $\mathbf{P}$ and $\mathbf{Q}$ over the dyads between $\mathcal{U}$ and $\mathcal{V}$. Here, "un-normalized" means that the

dyads are drawn uniformly with probabilities $1/N^2$, but the sum of probabilities is $|\mathcal{U}| \cdot |\mathcal{V}|/N^2$ and not 1. The unnormalized uniform distribution discourages the supremum inside the definition of $D_\square$ from choosing small blocks for maximizing (16). Note that normalized uniform distributions would lead $D_\square$ to choose small blocks, which do not reflect meaningful empirical estimates of the edge statistics (the edge densities of small blocks would not be interpretable as expected number of edges). To conclude, $D_\square(\mathbf{P}||\mathbf{Q})$ is interpreted as the maximal divergence between $\mathbf{P}$ and $\mathbf{Q}$ over all blocks, up to the best regularizers.

In our analysis we compute the log cut distance between the random decoded graph $\mathbf{P}$ and the deterministic target graph $\mathbf{A}$. While $\mathbf{A}$ has edge probabilities in $\{0,1\}$, $\mathbf{P}$ has edge probabilities in $[0,1)$ for Clam models. Therefore, we only require regularization for $\mathbf{A}$. We hence consider the following modified version of Definition 3.3.

**Definition 3.4.** Given an unweighted graph with adjacency matrix $\mathbf{A} \in \{0,1\}^{N \times N}$ and a random graph over the nodes set $[N]$, with independently Bernoulli distributed edges with probabilities $\mathbf{P} = \{0 \le p_{n,m} < 1\}_{n,m \in [N]}$, the *log cut distance* between $\mathbf{P}$ and $\mathbf{A}$ is defined to be

$D_\square(\mathbf{P}||\mathbf{A}) :=$

$\inf_{0 < d \le 1} \left( d + \frac{1}{N^2} \sup_{\mathcal{U},\mathcal{V} \subset [N]} \left| \log \Big( \prod_{n \in \mathcal{U}} \prod_{m \in \mathcal{V}} \frac{1 - p_{n,m}}{1 - (1-d)a_{n,m}} \Big) \right| \right).$

Lastly, in case both $\mathbf{P}$ and $\mathbf{Q}$ are $[0,1)$-valued, a simple version of the log cut distance is (15) with the choice $e = d = 0$, namely,

$$D_\square^0(\mathbf{P}||\mathbf{Q}) :=$$

$$\frac{1}{N^2} \sup_{\mathcal{U},\mathcal{V} \subset [N]} \left| \log \Big( \prod_{n \in \mathcal{U}} \prod_{m \in \mathcal{V}} \frac{1 - p_{n,m}}{1 - q_{n,m}} \Big) \right|. \qquad (17)$$

### 3.3. Universal Graph Autoencoders

We are now ready to define the universality of general pairwise autoencoders. Motivated by the fact that a Clam autoencoder is actually a family of autoencoders, parameterized by the number of communities, we also define general pairwise decoders as families.

**Definition 3.5.** A family of code spaces $\mathbb{R}^M$ and corresponding pairwaise decoders $D_M : \mathbb{R}^{2M} \to [0,1]$, parametrized by $M \in \mathbb{N}$, is called *universal* if for every $\epsilon > 0$ there is $M \in \mathbb{N}$ (which depends only on $\epsilon$) such that for every $N \in \mathbb{N}$ and every graph with adjacency matrix $\mathbf{A}$ and $N$ nodes there are $N$ points in the code space $\{z_n \in \mathbb{R}^M\}_{n=1}^N$ such that $D_\square(\mathbf{D}_M(\mathbf{z})||\mathbf{A}) < \epsilon$.

### 3.4. BigClam and PClam are Not Universal

We now show that BigClam (and hence also PClam) is not a universal autoencoder since it cannot approximate

bipartite graphs. Consider the bipartite graph $\mathbf{B}$ with $N$ nodes at each part, and probability $1 - e^{-a^2}$ for an edge between the two parts, and $0$ within each part. Since in this case all probabilities are less than 1, we can use (17) as the definition of the log cut distance. The analysis for Definition 3.4 extends naturally.

Let $\mathbf{P}$ be a decoded BigClam graph. We show that there is no way to make $D_\square^0(\mathbf{P}||\mathbf{Q})$ small by choosing the dimension $C$ uniformly with respect to $N$. In fact, we will show BigClam cannot approximate a bipartite graph at all.[2]

*Claim* 3.6. Under the above construction,

$$D_\square^0(\mathbf{P}||\mathbf{B}) \ge \frac{a^2}{16}.$$

As a result, BigClam is not a universal autoencoder.

The proof is given in Appendix A.1. We note that one can similarly show that BigClam is not universal also with respect to the log cut distance of Definition 3.4.

### 3.5. Universality of IeClam and PieClam

We are now ready to show that IeClam (and hence also PieClam) is a universal autoencoder. The proofs of the following two theorems are in Appendix A.3 and A.4.

We give two versions for the universality result. The first is without a cone restriction, and requires a relatively small number of communities for the given error tolerance. The corresponding decoder produces edge weights that can be negative. The second theorem is restricted to the pairwise cone of non-negativity, and has pessimistic asymptotics for the required number of communities given an error tolerance. The second decoder is guaranteed to produce proper edge probabilities in $[0,1)$.

**Theorem 3.7.** *For every epsilon $\epsilon > 0$, every $N \in \mathbb{N}$, and every adjacency matrix $\mathbf{A} \in [0,1]^{N \times N}$, there are $N$ affiliation features $\mathbf{F} \in \mathbb{R}^{2K}$ of dimension $K = -9 \log(\epsilon/2)^2/\epsilon^2$ such that the corresponding IeClam model $\mathbf{P} = \{P(n \sim m|\mathbf{f}_n, \mathbf{f}_m)\}_{n,m=1}^N$ satisfies $D_\square(\mathbf{P}||\mathbf{A}) < \epsilon$. Here, the log cut distance is from Definition 3.4. As a result, IeClam and PieClam are universal autoencoders with code space $\mathbb{R}^{2K}$.*

**Theorem 3.8.** *For every epsilon $\epsilon > 0$, every $N \in \mathbb{N}$, and every adjacency matrix $\mathbf{A} \in [0,1]^{N \times N}$, there are $N$ affiliation features $\mathbf{F}$ in the cone of pairwise non-negativity $\mathcal{T}^C \subset \mathbb{R}^{2C}$ of dimension $C = 2^{4\lceil -\log(\epsilon/2)^2/\epsilon^2 \rceil}$ such that the corresponding IeClam model $\mathbf{P} = \{P(n \sim m|\mathbf{f}_n, \mathbf{f}_m)\}_{n,m=1}^N$ satisfies $D_\square(\mathbf{P}||\mathbf{A}) < \epsilon$. Here, the log cut distance is from Definition 3.4. As a result, IeClam and PieClam are universal autoencoders with code space $\mathcal{T}^C$.*

---

[2]In Appendix A.2 we show that one can approximate a bipartite graph of $2N$ nodes using $C = N^2$ classes in BigClam if the model ignores self-loops.

### 3.6. Log cut distance and maximum likelihood

It is well known that it is NP hard to compute cut distance (Alon & Naor, 2006; Rohn, 2000), and hence also log cut distance. Instead, we observe that maximizing log likelihood, which is efficiently implemented with gradient ascent, leads to a small log cut distance in practice. See Figure 1.

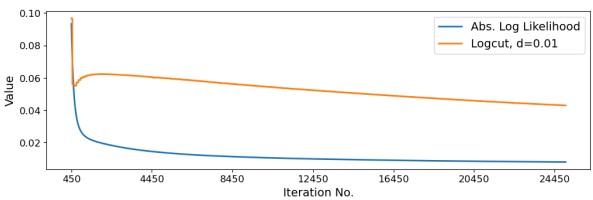

Figure 1: The value of log cut vs. the abosolute value of log likelihood along the optimization of IeClam on Squirrel. When log likelihood is optimized, log cut distance decreases.

## 4. Experiments

All of the experiments were run on Nvidia GeForce RTX 4090 and Nvidia L40 GPUs.[3]

### 4.1. Reconstructing Synthetic Priors

We consider a ground-truth synthetic prior $p$. We sample $N = 500$ points from $p$, decode the corresponding PieClam graph, and sample a simple graph from the random Bernoulli edges. Then, using the sampled graph as the target for the PieClam optimization, we fit PieClam on $\mathcal{T}^1$. The results are shown in Figure 2. We observe that the algorithm reconstructs both the positions of the nodes in the affiliation space and the original prior qualitatively well. Additional Experiments and results are given in Appendix D.4.

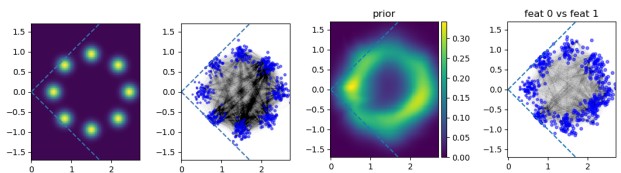

Figure 2: Left to right: Ground truth synthetic prior in $\mathcal{T}^1$; Graph sampled from the prior; Reconstructed prior via PieClam; Reconstructed nodes via PieClam.

### 4.2. Reconstructing Synthetic SBMs

In Figure 3 we consider a synthetic SBM with intercommunity connection probability of $0$ and intracommunity

connection probability of $0.5$. We sample a simple graph with $N = 210$ nodes from the SBM, and fit PClam and PieClam to is. The SBM is off diagonal, so it cannot be well approximated by the PClam model, while the PieClam model approximates it well qualitatively. Additional details are given in Appendix D.

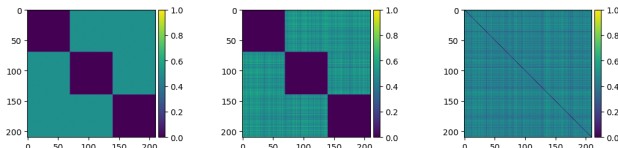

Figure 3: Left to right: Original SBM with 3 classes and 9 blocks; Adjacency matrix of the fitted PieClam graph, with two inclusive and two exclusive communities; Adjacency matrix of the fitted PClam graph, with four communities.

### 4.3. Anomaly Detection

In unsupervised node anomaly detection, one is given a graph with node features, where some of the nodes are unknown anomalies. The goal is to detect these anomalous nodes, without supervising on any example of normal or anomalous nodes, using only the structure of the graph and the node features.

We fit a Clam model to the graph and flag nodes as anomalous if they satisfy the following different criteria.

- **(S) Star probability:** Given any Clam model, a node $n$ is called anomalous if $\prod_{m \in \mathcal{N}(n)} P(n \sim m | \mathbf{F}) < \delta$.

- **(P) Prior probability:** Given any Clam model with prior, a node $n$ is called anomalous if $p(\mathbf{f}_n) < \delta$.

- **(PS) Prior star probability:** Given any Clam model with prior, a node $n$ is called anomalous if $p(\mathbf{f}_n) \prod_{m \in \mathcal{N}(n)} P(n \sim m | \mathbf{F}) < \delta$.

We reduce the dimension of the node features of the input graphs to $100$ using truncated SVD, unless the dimension of the features is smaller than $100$ in which case we only normalize them to have zero mean and standard deviation one. We use an affiliation space embedding dimension of $30$ for $\mathbf{F}$ for PieClam and IeClam, and $24$ for BigClam. Every Clam method starts with a random embedding $\mathbf{F}$. Clam models with prior are trained with the following steps. ($\mathbf{F}$-$t$): given a fixed prior $p$, optimize only $\mathbf{F}$ for $t$ steps. ($p$-$t$): given a fixed embedding $\mathbf{F}$, optimize only $p$ for $t$ steps. For regularization, when training, in each iteration of $p$ we add Gaussian noise with STD $0.05$ to the affiliation features before plugging them into the prior. We train PieClam with the scheme $\mathbf{F}$-$500 \rightarrow p$-$1300 \rightarrow \mathbf{F}$-$500 \rightarrow p$-$1300 \rightarrow \mathbf{F}$-$500 \rightarrow p$-$1300$ with learning rate of $2e^{-6}$ on $\mathbf{F}$ and $1e^{-6}$ on $p$. In

---

[3]Our code can be found in: https://github.com/danizil/PieClam

| Method | Reddit | Elliptic | Photo |
|---|---|---|---|
| (S)- IeClam | **64.1** | 43.6 | 57.7 |
| (S) - PieClam | *64.0 | 43.5 | 59.0 |
| (P) - PieClam | 46.8 | **63.2** | 45.7 |
| (PS) - PieClam | 64.0 | 53.8 | **59.0** |
| (S) - BigClam | 63.7 | 43.4 | *58.1 |
| DOMINANT | 51.1 | 29.6 | 51.4 |
| AnomalyDAE | 50.9 | *49.6 | 50.7 |
| OCGNN | 52.5 | 25.8 | 53.1 |
| AEGIS | 53.5 | 45.5 | 55.2 |
| GAAN | 52.2 | 25.9 | 43.0 |
| TAM | 60.6 | 40.4 | 56.8 |

Table 1: Comparison of Clam anomaly detectors with competing methods. First place in **boldface**, second with underline, third with *star. The accuracy metric is areas under curve (AUC).

every alternation between two $(p - t)$ and $(\mathbf{F} - \mathbf{t})$ schemes, both learning rates are decreased by a factor of 2. For the models which only optimize $\mathbf{F}$ (IeClam and BigClam) we use the following configurations. We train IeClam with 2500 iterations with learning rate 1e-6. We train BigClam with 2200 iterations with learning rate 1e-6. More details on hyper-parameters are given in Appendix D.2, and in our GitHub repository.

In Table 1 we compare the performance of Clam methods to DOMINANT (Ding et al., 2019a), AnomalyDAE (Fan et al., 2020), OCGNN (Wang et al., 2021), AEGIS (Ding et al., 2021), GAAN (Chen et al., 2020b) and TAM (Qiao & Pang, 2024) on the datasets Reddit, Elliptic (Elliptic) (Weber et al., 2019), and Photo (Shchur et al., 2018a). The hyper-parameters of the competing methods are taken as the recommended values from the respective papers. The results are taken from Table 1 in (Qiao et al., 2024). We observe that our methods are the first and second place on all datasets. Moreover, (PS)- PieClam beats the competing methods in all three datasets.

### 4.4. Link Prediction

In supervised link prediction, one is given a graph where for some of the dyads it is unknown if they are edges or not. The goal is to predict the connectivity of the omitted dyads. Given a Clam model fitted to the known data, we predict the probability of an edge using the conditional probability given in (7). In our experiments, the prior and features are not directly used for prediction, but they are used to train the Clam model if it has a prior. In Appendix D.3 we show that the log likelihood, restricted to the known dyads, can

be efficiently computed by

$$2\hat{l}(\mathbf{F}, E, \check{\mathcal{E}}) = 2 \sum_{n \in [N]} \log(p(\mathbf{f}_n)) + \sum_{(n,m) \in E \setminus \check{E}} \log(1 - e^{-\mathbf{f}_n^\top \mathbf{L} \mathbf{f}_m})$$
$$- \sum_{n \in [N]} \mathbf{f}_n^\top \sum_{m \in [N]} \mathbf{L} \mathbf{f}_m + \sum_{(n,m) \in E \dot\cup \check{\overline{E}} \dot\cup D} \mathbf{f}_n^\top \mathbf{L} \mathbf{f}_m$$

where $E$ is the set of edges, $\check{E}$ is the set of omitted edges, $\check{\overline{E}}$ is the set of omitted non-edges, $\check{\mathcal{E}} = \check{E} \dot\cup \check{\overline{E}}$ is the set of omitted dyads, and $D = \{(n, n) | n \in [N]\}$. This computation is as efficient as message passing when the number of omitted non-edges is of the same order as $E$. After assigning probabilities to all of the omitted dyads, we calculate the AUC score of the classification. We use the experimental setting presented in (Zhou et al., 2022).

For each dataset, we generate 10 random splits into a test set containing $10\%$ of the edges, along with 5 randomly sampled non-edges for every omitted edge, and the rest is the training set. For each split and each configuration of the hyperparameters, we generate three random validation sets consisting of $5\%$ of the edges from the training set and 5 non-edges per omitted edge also from the training set, leaving the rest of the original training set for training. We perform hyperparameter optimization as follows: Each hyperparameter configuration is trained against each of its respective 30 training sets, and its accuracy is evaluated over the corresponding 30 validation sets and averaged to obtain one mean validation accuracy per configuration. We then choose the configuration with maximal mean validation accuracy. The model with these optimal hyperparameters is then trained on each of the original 10 training sets and tested on each of the corresponding test sets, 10 times for each split. We report the mean accuracy across the 100 trained models on the 10 test sets, along with the standard deviation of the mean. We run our experiments on the datasets Squirrel and Texas (Pei et al., 2020), Photo (Shchur et al., 2018b) and Facebook100's Johns Hopkins (Lim et al., 2021b), and compare PieClam to the baselines AA (Adamic & Adar, 2003), VGAE (Kipf & Welling, 2016), GAT (Velickovic et al., 2017), LINKX (Lim et al., 2021a) and Disenlink (Zhou et al., 2022). The results are presented in Table 2 (and with error bars in Table 6), and the baseline results are taken from Table 2 in (Zhou et al., 2022). An in depth explanation on the experimental setup and a table with confidence intervals is given in Appendix D.3.

We conducted additional link prediction experiments on the **OGB-DDI** dataset. In this dataset, nodes represent different drugs, and an edge between two nodes indicates that a different effect occurs when the drugs are taken together versus separately. Our model achieves state-of-the-art performance on this benchmark in terms of the **Hits@20** metric (using OGB's official evaluator). A comparison between our model (**IeClam**) and selected baselines on the dataset's

| Method | Squirrel | Photo | Texas | JH55 |
|---|---|---|---|---|
| PieClam | **98.7** | **98.4** | **85.0** | 95.5* |
| BigClam | 98.5 | 97.4 * | 78.2 * | 94.9 |
| VGAE | 98.2 | 94.9 | 68.6 | 92.8 |
| GAT | 98.0 | 97.3 | 68.5 | 94.3 |
| LINKX | 98.1 | 97.0 | 75.8 | 93.4 |
| AA | 97.1 | 97.4 | 53.1 | 96.1 |
| DisenLink | 98.3* | 97.9 | 81.0 | **97.5** |

Table 2: Comparison of Clam link predictors with competing models. First place in **boldface**, second with underline, third with *star

| Model | Hits@20 (Test) | AUC (Test) |
|---|---|---|
| **IeClam** | **88.10** $\pm$ **1.70** | 99.84 $\pm$ 0.01 |
| NCN | 76.52 $\pm$ 10.47 | **99.97** $\pm$ **0.00** |
| GraphSAGE | 49.84 $\pm$ 15.56 | 99.96 $\pm$ 0.00 |
| GCN | 49.90 $\pm$ 7.23 | 99.86 $\pm$ 0.03 |
| BUDDY | 29.60 $\pm$ 4.75 | 99.81 $\pm$ 0.02 |
| Node2Vec | 34.69 $\pm$ 2.90 | 99.78 $\pm$ 0.04 |

Table 3: Performance on the OGB-DDI dataset. First place is in **boldface**.

test set is presented in Table 3. Benchmark results are taken from (Zhang et al., 2022; Li et al., 2023; Tan et al., 2022).

## 5. Conclusion

We introduced PieClam, a new probabilistic graph generative model. PieClam models graphs via embedding the nodes into an inclusive and exclusive communities space, learning a prior distribution in this space, and decoding pairs of points in this space to edge probabilities, such that points are more likely to be connected the more inclusive communities and the less exclusive communities they share. We showed that PieClam is a universal autoencoder, able to approximate any graph, where the budget of parameters per node (the number of communities) can be predefined, irrespective of any property of a specific graph, not even the number of nodes. Our experiments show that PieClam achieves competitive results when used in graph anomaly detection.

One limitation of PieClam is that, for attributed graphs, it only models the node features through the prior in the community affiliation space, but not via the conditional probabilities of the edges (given the community affiliations). Future work will deal with extending PieClam to also include the node (or edge) features in the edge conditional probabilities. Another limitation is in the analysis. While our methods performs well for sparse graphs, our analysis involving the log cut distance is mainly appropriate for dense graphs. Indeed, the term $1/N^2$ in (16) leads all sparse graphs with $|E| \ll N^2$ to be trivially close to each other. Future work will extend our log cut similarity

measure to sparse graphs. This can be done, e.g., similarly to the sparse theory in (Finkelshtein et al., 2024), or using stretched graphons (Borgs et al., 2018), $L^p$ graphons (Borgs et al., 2014), graphings (Lovász, 2012) or graphops (Backhausz & Szegedy, 2018).

## Impact Statement

This paper presents work whose goal is to advance the field of Machine Learning. There are many potential societal consequences of our work, none which we feel must be specifically highlighted here.

## Acknowledgments

This research was supported by a grant from the United States-Israel Binational Science Foundation (BSF), Jerusalem, Israel, and the United States National Science Foundation (NSF), (NSF-BSF, grant No. 2024660), and by the Israel Science Foundation (ISF grant No. 1937/23).

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

# Supplementary Material

## A. Proofs

### A.1. Proof That BigClam Is Not Universal

In this subsection we Prove Claim 3.6. Consider the bipartite graph $\mathbf{B}$ with $N$ nodes at each part, and probability $1 - e^{-a^2}$ for an edge between the two parts, and 0 within each part. Consider (17) as the definition of the log cut distance. Let $\mathbf{P}$ be a decoded BigClam graph from the affiliation features $\mathbf{F}$.

Denote by $\tilde{\mathbf{P}}$ the matrix with entries

$$\tilde{p}_{n,m} = -\log(1 - p_{n,m}) = \mathbf{f}_n^\top \mathbf{f}_m,$$

and by $\tilde{\mathbf{B}}$ the matrix with entries $\tilde{b}_{n,m} = -\log(1 - b_{n,m})$. We show that there is no way to make $\|\tilde{\mathbf{P}} - \tilde{\mathbf{Q}}\|_\square$ small.

*Claim* A.1. Under the above construction,

$$D_\square^0(\mathbf{P}||\mathbf{B}) \geq \frac{a^2}{16}.$$

As a result, BigClam is not a universal autoencoder.

*Proof.* Note that $\tilde{b}_{n,m} = a^2$ if $n, m$ are in opposite parts, and $\tilde{b}_{n,m} = 0$ if $n, m$ are on the same side. We index the nodes such that $[N]$ is the first side of the graph, and $[N] + N$ is the second side. For $n \in [N]$ we denote $\mathbf{q}_n = \mathbf{f}_n$ and $\mathbf{y}_n = \mathbf{f}_{n+N}$. Next, we use the identity

$$D_\square^0(\mathbf{P}||\mathbf{Q}) = \|\tilde{P} - \tilde{B}\|_\square,$$

and bound the right-hand-side from below.

First, by the definition of cut norm, for every $\mathcal{U}, \mathcal{V} \subset [2N]$,

$$\|\tilde{P} - \tilde{B}\|_\square \geq \left| \frac{1}{4N^2} \sum_{n \in \mathcal{U}} \sum_{m \in \mathcal{V}} (\tilde{p}_{n,m} - \tilde{b}_{n,m}) \right|.$$

Hence, for $\mathcal{U}_1 = \mathcal{V}_1 = [N], \mathcal{U}_2 = \mathcal{V}_2 = [N] + 1, \mathcal{U}_3 = [N], \mathcal{V}_3 = [N] + N$, and $\mathcal{U}_4 = [N] + N, \mathcal{V}_4 = [N]$, we have

$$\|\tilde{P} - \tilde{B}\|_\square \geq$$

$$\frac{1}{16N^2} \sum_{j=1}^4 \left| \sum_{n \in \mathcal{U}_j} \sum_{n \in \mathcal{V}_j} (\tilde{p}_{n,m} - \tilde{b}_{n,m}) \right|$$

$$= \frac{1}{16N^2} \sum_{n=1}^N \sum_{m=1}^N \mathbf{q}_n^\top \mathbf{q}_m + \frac{1}{16N^2} \sum_{n=1}^N \sum_{m=1}^N \mathbf{y}_n^\top \mathbf{y}_m$$

$$+ \frac{1}{8N^2} \sum_{n=1}^N \sum_{m=1}^N (a^2 - \mathbf{q}_n^\top \mathbf{y}_m). \tag{18}$$

Denote

$$\mathbf{q} = \frac{1}{N} \sum_{n=1}^N \mathbf{q}_n, \quad \mathbf{y} = \frac{1}{N} \sum_{n=1}^N \mathbf{y}_n.$$

With these notations, (18) can be written as

$$\|\tilde{P} - \tilde{B}\|_\square \geq$$
$$\frac{1}{16} \mathbf{q}^\top \mathbf{q} + \frac{1}{16} \mathbf{y}^\top \mathbf{y} + \frac{1}{8}(a^2 - \mathbf{q}^\top \mathbf{y})$$
$$= \frac{1}{16} \left( a^2 + (\mathbf{q}^\top - \mathbf{y}^\top)(\mathbf{q} - \mathbf{y}) \right) \geq \frac{a^2}{16}.$$

$\square$

We note that one can similarly show that BigClam is not universal also with respect to the log cut distance of Definition 3.4.

## A.2. BigClam With No Self Loops Approximating Bipartite Graphs

Consider the above bipartite graph $\mathbf{B}$ with $N$ nodes at each part, and probability $1 - e^{-a^2}$ for an edge between the two parts, and $0$ within each part. If we redefine the BigClam decoder to have no self-loops, namely, $\mathbf{P}$ has entries $p_{n,m} = P(n \sim m | \mathbf{f}_n, \mathbf{f}_m)$ for $n \neq m$, and $p_{n,m} = 0$ for $n = m$, then one can obtain a bipartite $\mathbf{P}$ with $C = N^2$ communities as follows.

In the following analysis, an addition or multiplication of a set by a scalar is defined to be the addition or multiplication of every element in the set by this scalar. Encode each node $n \in [N]$ in part 1 to $\mathbf{f}_n$ with $f_n^c = a$ for $c \in [N] + n(N - 1)$ and $f_n^c = 0$ otherwise. Encode every node $n \in [N] + N$ from side 2 to $\mathbf{f}_n$ with $f_n^c = a$ for $c \in N([N] - 1) + n$ and $f_n^c = 0$ otherwise. It is easy to see that the corresponding $\mathbf{P}$ is bipartite with edge probability between the parts being $1 - e^{-a^2}$.

## A.3. Proof of the Universality of PieClam

The proof is based on a version of the weak regularity lemma for intersecting communities. While the standard weak regularity lemma (Frieze & Kannan, 1999; Lovász & Szegedy, 2007) partitions the graph into disjoint communities, it is well known that allowing the communities to overlap allows using much less communities, which improves the asymptotics of the approximation. The regularity lemma was used in the context of graph machine learning in (Levie, 2023; Finkelshtein et al., 2024). To formalize the relevant version of the weak regularity theorem for our analysis, we first need to cite a definition from (Finkelshtein et al., 2024).

**Definition A.2.** A (hard) intersecting community graph (ICG) with $N$ nodes and $K$ communities is a matrix $\mathbf{C} \in \mathbb{R}^{N \times N}$ of the following form. There exist $K$ column vectors $\mathbf{Q} = \left(\mathbf{q}_k \in \{0, 1\}^N\right)\right)_{k=1}^K \in \{0, 1\}^{N \times K}$ and $k$ coefficients $\mathbf{r} = (r_k \in \mathbb{R})_{k=1}^K \in \mathbb{R}^K$ such that
$$\mathbf{C} = \mathbf{Q}\text{diag}(\mathbf{r})\mathbf{Q}^\top.$$

The following is a special case of the weak regularity lemma from (Finkelshtein et al., 2024), up to the small modification to the adjacency matrix, allowing it to have values in $[0, R]$ instead of $[0, 1]$.

**Theorem A.3.** *Let $\mathbf{A} \in [0, R]^{N \times N}$ be an adjacency matrix of a graph with $N$ nodes. Let $\epsilon > 0$. Denote $K = \frac{9R^2}{4\epsilon^2}$. Then, there exists a hard ICG $\mathbf{C}$ with $K$ communities such that*

$$\|\mathbf{A} - \mathbf{C}\|_\square \leq \epsilon. \tag{19}$$

*Proof of Theorem 3.7.* Let $\epsilon > 0$. Let $\mathbf{A} \in [0, 1]^{N \times N}$ be an adjacency matrix. Let $0 < d \leq 1$.

Consider the matrix $\tilde{\mathbf{A}}$ with entries
$$\tilde{a}_{n,m} = -\log(1 - (1 - d)a_{n,m}).$$

In the following construction, we build IeClam affiliation features $\mathbf{F}$ and we want
$$1 - \exp(-\mathbf{f}_n^\top \mathbf{L}\mathbf{f}_m) \approx (1 - d)a_{n,m}.$$

Note that $-\log(1 - (1 - d)a_{n,m})$ is increasing in $(1 - d)a_{n,m}$. For $(1 - d)a_{n,m} = 0$ the value of this function is 0, and for $(1 - d)a_{n,m} = 1 - d$ it is $R = -\log(d)$. Choose $d = \epsilon/2$. For this specific choice of $d$, if we replace $d$ by $\epsilon/2$ in the definition of $D_\square$ and omit the infimum, we get an upper bound of $D_\square(\mathbf{P}, \mathbf{A})$.

Using the overlaping weak regularity lemma, we approximate $\tilde{\mathbf{A}}$ by an ICG with
$$K = \frac{-9\log(\epsilon/2)^2}{\epsilon^2}$$

communities,
$$\mathbf{C} = \mathbf{Q}\text{diag}(\mathbf{r})\mathbf{Q}^\top,$$

such that
$$\|\tilde{\mathbf{A}} - \mathbf{C}\|_\square \leq \epsilon/2.$$

Let $\mathbf{r}_+ = \text{ReLU}(\mathbf{r})$ and $\mathbf{r}_- = \text{ReLU}(-\mathbf{r})$. Denote
$$\mathbf{C}_+ = \mathbf{Q}\text{diag}(\sqrt{\mathbf{r}_+})$$

and

$$\mathbf{C}_- = \mathbf{Q}\mathrm{diag}(\sqrt{\mathbf{r}_-}).$$

Denote the rows of $\mathbf{C}_+$ by $\mathbf{t}_n$ and the rows of $\mathbf{C}_-$ by $\mathbf{s}_n$, for $n = 1, \ldots, N$. For each $n \in [N]$ we concatenate $(\mathbf{t}_n, \mathbf{s}_n)$ to define the affiliation feature $\mathbf{f}_n$, with the first $K$ coordinates being the inclusive communities, and the last $K$ coordinates being the exclusive communities. Denote the corresponding IeClam matrix by $\mathbf{P}$.

It is easy to see that $\mathbf{f}_n^\top \mathbf{L} \mathbf{f}_m$ is the $(n, m)$ entry of $\mathbf{Q}\mathrm{diag}(\mathbf{r})\mathbf{Q}^\top$. This also proves that

$$\| - \log(1 - (1 - \epsilon)\mathbf{A}) + \log(1 - \mathbf{P})\|_\square$$

$$= \|\tilde{\mathbf{A}} - \mathbf{C}\|_\square \le \epsilon/2.$$

We can now summarize

$$D_\square(\mathbf{P}\|\mathbf{A}) \le$$

$$\epsilon/2 + \frac{1}{N^2} \sup_{\mathcal{U},\mathcal{V} \subset [N]} \Big| \log \Big( \prod_{n \in \mathcal{U}} \prod_{m \in \mathcal{V}} \frac{1 - p_{n,m}}{1 - (1 - \epsilon/2)a_{n,m}} \Big) \Big|$$

$$= \epsilon/2 + \frac{1}{N^2} \sup_{\mathcal{U},\mathcal{V} \subset [N]} \Big| \sum_{n \in \mathcal{U}} \sum_{m \in \mathcal{V}} \Big( - \log \big(1 - (1 - \epsilon)a_{n,m}\big)$$

$$+ \log \big(1 - p_{n,m}\big) \Big) \Big|$$

$$= \epsilon/2 + \|\tilde{\mathbf{P}} - \mathbf{C}\|_\square = \epsilon.$$

$\square$

### A.4. Proof of the Universality of IeClam in the Pairwise Cone of Non-negativity

For this result, we use the standard weak regularity lemma for non-intersecting classes. It is based on the weak regularity lemma from (Frieze & Kannan, 1999; Lovász & Szegedy, 2007), see also Lemma 9.3 and Corollary 9.13 from (Lovász, 2012).

**Definition A.4.** A *block matrix* $\mathbf{B}$ with $K$ classes is a symmetric matrix $\mathbf{B} \in [0, \infty)^{N \times N}$ for which there exists a partition of $[N]$ into $K$ disjoint sets, called *classes*, $\mathcal{C}_1, \ldots, \mathcal{C}_K$ (with $\cup \mathcal{C}_j = [N]$), such that for every pair of classes $i, j \in [K]$, there is a constant $c_{i,j} \ge 0$ such that $b_{n,m} = c_{i,j}$ for any two nodes $n \in \mathcal{C}_i$ and $m \in \mathcal{C}_j$.

**Theorem A.5.** *Let $\mathbf{A} \in [0, R]^{N \times N}$ be an adjacency matrix of a graph with $N$ nodes. Let $\epsilon > 0$. Denote $K = 2^{2\lceil R^2/\epsilon^2 \rceil}$. Then, there exists a block matrix $\mathbf{B}$ with $K$ (disjoint) classes such that*

$$\|\mathbf{A} - \mathbf{B}\|_\square \le \epsilon. \tag{20}$$

We stress that Theorem A.5 guarantees non-negative block values of $\mathbf{B}$, while in Theorem A.3, in general, the matrix $\mathbf{C}$ may have negative entries.

*Proof of Theorem 3.8.* We start similarly to the proof of Theorem 3.7. Let $\epsilon > 0$. Let $\mathbf{A} \in [0, 1]^{N \times N}$ be an adjacency matrix. Consider the matrix $\tilde{\mathbf{A}} \in [0, -\log(\epsilon/2)]$ with entries

$$\tilde{a}_{n,m} = -\log(1 - (1 - \epsilon/2)a_{n,m}).$$

In the following, we build IeClam affiliation features $\mathbf{F}$ such that

$$1 - \exp(-\mathbf{f}_n^\top \mathbf{L} \mathbf{f}_m) \approx (1 - \epsilon/2)a_{n,m}.$$

By the weak regularity lemma (Theorem A.5), we approximate there is a non-negative block matrix $\mathbf{B}$ with $K = 2^{2\lceil -\log(\epsilon/2)^2/\epsilon^2 \rceil}$ classes $\mathcal{C}_1, \ldots, \mathcal{C}_K$, such that

$$\|\tilde{\mathbf{A}} - \mathbf{B}\|_\square \le \epsilon.$$

We now take the affiliation space to have $C = K^2$ inclusive communities and $C = K^2$ exclusive communities.

For each $n \in [N]$, let $k_n$ be the class such that $c \in \mathcal{C}_{n_k}$. For each pair of classes $i, j \in [K]$, let $c_{i,j} \geq 0$ denote the edge weight between $\mathcal{C}_i$ and $\mathcal{C}_j$.

The feature of each $n \in [N]$ at the inclusive channel $c = (K-1)k_n + k_n$ is $t_n^c = \sqrt{c_{k_n,k_n}}$. It is $t_n^c = \sqrt{c_{k_n,j}/4}$ at inclusive channels $c = (K-1)k_n + j$ and $c = (K-1)j + k_n$, and $s_n^c = -\sqrt{c_{k_n,j}/4}$ at exclusive channels $c = (K-1)k_n + j$ and $c = (K-1)j + k_n$. In all other channels $t_n^c$ and $s_n^c$ are zero. Note that $\mathbf{f}_n$ belongs to the cone of pairwise non-negativity $\mathcal{T}^{K^2} \subset \mathbb{R}^{2K^2}$.

It is now direct to see that $\mathbf{f}_n^\top \mathbf{L} \mathbf{f_m} = c_{k_n,k_m}$. As a result, as in the proof of Theorem 3.7, we get

$$D_\square(\mathbf{P}||\mathbf{A}) \leq$$

$$\epsilon/2 + \frac{1}{N^2} \sup_{\mathcal{U},\mathcal{V}\subset[N]} \Big| \log\Big(\prod_{n\in\mathcal{U}}\prod_{m\in\mathcal{V}} \frac{1-p_{n,m}}{1-(1-\epsilon/2)a_{n,m}}\Big)\Big|$$

$$= \epsilon/2 + \frac{1}{N^2} \sup_{\mathcal{U},\mathcal{V}\subset[N]} \Big| \sum_{n\in\mathcal{U}}\sum_{m\in\mathcal{V}} \Big( -\log\big(1-(1-\epsilon)a_{n,m}\big)$$

$$+ \log\big(1-p_{n,m}\big)\Big)\Big|$$

$$= \epsilon/2 + \|\tilde{\mathbf{P}} - \mathbf{B}\|_\square = \epsilon.$$

$\square$

# B. Extended Related Work

## B.1. Message Passing Algorithms and Networks

The message passing algorithm is a general architecture for processing graph-signals. An MPNN operates on the graph data by aggregating the features in the neighborhood of each node, allowing for information to travel along the edges. The first example of this scheme (Message Passing) was originally suggested by Pearl et al (Pearl, 1982), and was combined with a neural network in (Duvenaud et al., 2015), and later generalized in (Gilmer et al., 2017b). Most graph neural networks applied in practice are specific instances of MPNNs (Gilmer et al., 2017a; Kipf & Welling, 2017; Velickovic et al., 2017).

In MPNNs, information is exchanged between nodes along the graph's edges. Each node combines the incoming messages from its neighbors using an *aggregation scheme*, with common methods being summing, averaging, or taking the coordinate-wise maximum of the messages. Let $T \in \mathbb{N}$ represent the number of layers, and define two sequences of positive integers $(c_t)_{t=0}^T$ and $(d_t)_{t=0}^T$ representing the feature dimensions in the hidden layers $\{\mathcal{G}_t\}_{t=0}^T = \{\mathbb{R}^{c_t}\}_{t=0}^T$ and $\{\mathcal{S}_t\}_{t=0}^T = \{\mathbb{R}^{d_t}\}_{t=0}^T$. Define the message functions as $M^t : \mathcal{S}_t \times \mathcal{S}_t \to \mathcal{G}_t$ and unpdate functions as $U^t : \mathcal{G}_t \times \mathcal{S}_t \to \mathcal{S}_{t+1}$. The features $\mathbf{f}_n^{t+1} \in \mathcal{S}_{t+1}$ at layer $t + 1$ of the nodes $n \in [N]$ are computed from the features $\mathbf{f}_m^t \in \mathcal{S}_t$ by

$$\mathbf{m}_n^t = \sum_{k\in\mathcal{N}(n)} M^t(\mathbf{f}_n^t, \mathbf{f}_k^t)$$

$$\mathbf{f}_n^{t+1} = U^t(\mathbf{m}_n^t, \mathbf{f}_n^t).$$

Here, $M^t$, $U^t$ and $m^t$ are called the message function, update function and mail at time $t$ respectively. The summation over the messages can also be replaced for any node by any function $Agg_n^t : \prod_{|\mathcal{N}(n)|} \mathcal{G}_t \to \mathcal{G}_t$ which is permutation invariant.

At step $T$ a readout space can be defined $\mathcal{R}_T = \mathbb{R}^{b_T}$, with a permutation invariant readout function $R^T$, e.g., summing, averaging, or taking the max of all of the nodes of the graph. This produces a vector representation of the whole graph.

## B.2. Deep Generative Models

Generative models in machine learning assume that training data is generated by some underlying probability distribution. One goal in this context is to approximate this distribution, or build a model that approximates a random sampler of data points from this distribution. Hence, generative models can be used to generate synthetic data, mimicking training data by

sampling from the distribution (Harshvardhan et al., 2020; Dinh et al., 2016; Mohamed & Lakshminarayanan, 2016), or to infer the probability of unseen data by substituting it into the probability function. The latter can be useful in tasks like anomaly detection (Pang et al., 2021) in which the model can asses whether a sample is probable under the learned model. Two examples of generative models are Generative Adversarial Networks (GANs) (Goodfellow et al., 2014; Radford, 2015) and Variational Autoencoders (VAEs) (Kingma, 2013), which are used both for inference and generation.

VAE models consist of an encoder and a decoder. The encoder maps data from a high-dimensional *data space* to a lower-dimensional, simpler, *code space*. The decoder reverses this process, transforming data from the code space back into the data space. The code space serves as a bottleneck, capturing the essential features of the training data, which is often high-dimensional (e.g., images, social network graphs) and thus more complex than the code space. If the model trains by encoding followed by decoding, then minimizing the difference between the input and its decoded version, it is called an *autoencoder*.

In a VAE, training involves encoding each data point to a known distribution (typically Gaussian), sampling from this distribution, decoding it, and then minimizing the difference between the distribution of the original data and the decoded data. For a survey on VAEs, see (Tschannen et al., 2018).

### B.3. Graph Generative Models

Graph generative models learn a probability distributions of graphs. Such models allow for various tasks where the goal is not only to analyze existing graphs but also to predict or simulate new graph data.

Some classical generative models are pre-defined probabilistic models, e.g., the Erdős–Rényi model, Preferential attachment, Watts–Strogatz model, and more. See (Newman, 2003) for a review. Other graph generative models are learned from data, e.g., (Lee & Wilkinson, 2019; Kipf & Welling, 2016; You et al., 2018; Bojchevski et al., 2018).

Applications of generative graph models include social network analysis (Wang et al., 2018; Grover et al., 2019; Harshvardhan et al., 2020). anomaly detection B.6, graph synthesis (You et al., 2018; Guo & Zhao, 2022), data augmentation (Ding et al., 2019b), and protein interaction modeling (e.g. for drug manufacturing) (De Cao & Kipf, 2018; Ingraham et al., 2019), to name a few. For a review see (Ma et al., 2021).

**Deep Graph Autoencoders.** In graph deep learning-based autoencoders, one estimates the data distribution by learning to embed the nodes to a code space, in which the data distribution is defined to be some standard distribution, e.g., Gaussian, in such a way that the encoded nodes can be recunstructed back to the graph with small error. Graph VAEs (Kipf & Welling, 2016; Grover et al., 2019; Samanta et al., 2020; Mehta et al., 2019) embed the data into the code space by minimizing the evidence lower bound loss comprised of the decoding loss and the KL divergence between the encoded distribution and a Gaussian prior.

**GAN-based Graph generative models.** In (Wang et al., 2018), a GAN method for graphs generates a neighborhood for each of the nodes and the discriminator gives a probability score for each edge. This method also formulate a graph version of softmax, and offer a random walk based generating strategy. Another GAN model that is used for anomaly detection is GAAN (Chen et al., 2020b). In GAAN, the ground truth and generated node attributes are encoded into a latent space from which the adjacency between any two nodes is decoded using a sigmoid of the inner product between their latent features.

**Normalizing Flows-based Graph Models.** Normalizing flows models (Dinh et al., 2016) also have adaptations for graph data. For example, the work of (Liu et al., 2019; Madhawa et al., 2019) offers a version of coupling blocks that use message passing neural networks. See more details on normalizing flows in Section B.8.

### B.4. Stochastic block models

A stochastic block model (SBM) is a generative model for random graphs. A basic stochastic block model is defined by specifying a number of *classes* $K$, the probability $p_k$ of a random node being in block $k$, for $k \in [K]$, where $\sum_k p_k = 1$, and an array of values $\mathbf{C} = \{c_{k,l}\}_{k,l=1}^J \in [0,1]^{J \times K}$ indicating edge probabilities between classes. Each node of a randomly generated graph of $N$ nodes is independently chosen to belong to one of the classes at random, with probabilities $\{p_k\}_{k \in [K]}$. Then, the edges of the graph are chosen independently at random according to the following rule. For each $n \in [N]$, denote by $k_n \in [K]$ the class of $n$. Each dyad $(n, m) \in [N]^2$ is chosen to be an edge in probability $c_{k_n, k_m}$. Namely, the entries $a_{n,m}$ of the adjacency matrix of the random graph are independent random Bernoulli variables. See (Lee & Wilkinson,

2019) for a review on SBMs.

For each node $n$, denote by $\mathbf{f}_n \in \{0,1\}^K$ the vector such that $f_n^c = 1$ if and only if $k_n = c$. Hence, in a basic SBM, the presence of an edge between nodes $n$ and $m$ follows a Bernoulli distribution with parameter $P(n \sim m | \mathbf{f}_m, \mathbf{f}_n, \mathbf{C}) = \mathbf{f}_n^\top \mathbf{C} \mathbf{f}_m$, where the adjacency matrix is $\mathbf{P} = \mathbf{F} \mathbf{C} \mathbf{F}^\top$, where $\mathbf{F} \in \mathbb{R}^{N \times K}$ is the matrix where each row $n$ has the feature $\mathbf{f}_n$ (Nowicki & Snijders, 2001b). This model can be extended to intersecting classes, where now $\mathbf{f}_n$ can have more than one nonzero entry (Mørup et al., 2011; Miller et al., 2009; Palla et al., 2012).

In a *Bernouli-Poisson SBM* (Yang & Leskovec, 2013; Zhou, 2015; Shchur & Günnemann, 2019), the probability for a non-edge is modeled by a Poisson distribution. The idea is that the more classes $n$ and $m$ share, the higher the probability that there is an edge between them. Hence,

$$P(n \sim m | \mathbf{f}_m, \mathbf{f}_n, \mathbf{C}) = 1 - e^{-\mathbf{f}_n^\top \mathbf{C} \mathbf{f}_m}, \tag{21}$$

with the expected number of edges being $\mathbf{f}_n^\top \mathbf{C} \mathbf{f}_m$.

In both the Bernouli and Bernouli-Poisson models, the probabilisic model of the entire graph is given by a product of the probabilities of all of the events

$$P(E | \mathbf{F}, \mathbf{C}) = \sqrt{\prod_{n \in [N]} \left( \prod_{m \in \mathcal{N}(n)} P(n \sim m) \prod_{m \notin \mathcal{N}(n)} P(\neg(n \sim m)) \right)}.$$

Here, the square root is taken since we assume that the graph is undirected so the product goes over all of the edges twice (Aicher et al., 2015; Mørup et al., 2011).

When fitting an SBM to a graph, both the class affiliations of nodes and the block structure $\mathbf{C}$ are learned (Snijders & Nowicki, 1997; Nowicki & Snijders, 2001a; Latouche et al., 2012).

### B.5. Community Affiliation Models

Community detection is a fundamental task in network analysis, aiming to identify groups of nodes that are more densely connected internally than with the rest of the network. This process is useful for understanding the structure and function of complex networks, such as social, biological, and information networks (Fortunato & Hric, 2016).

Although there exist models in which each node belongs to only one community as in traditional SBM models (Holland et al., 1983), and some relatively new deep learning models such as (Cavallari et al., 2017), it was shown that real-world networks often exhibit overlapping communities (Yang & Leskovec, 2014; 2013), where nodes belong to multiple groups and the probability of connectivity increases the more communities two nodes share. This indeed makes intuitive sense when looking at, e.g., social networks, where the more common interests and social circles people share the more they are likely to connect.

An example of a community affiliation model that can generate new graphs is AGM (Yang & Leskovec, 2012), which classifies all of the nodes in a graph into several communities, where each community $k$ has a probability $p_k$ for two member nodes to connect. If we denote by $C_{n,m}$ the set of communities two nodes $n, m \in [N]$ share, then the probability of an edge between $n$ and $m$ is

$$p(n \sim m) = 1 - \prod_{k \in C_{nm}} (1 - p_k).$$

To sample a new graph from a trained model, nodes are sampled and assigned communities based on the relative sizes of communities in the training graph, and edges are connected based on their mutual community memberships.

Community affiliations can also be continuous, where nodes have varying degrees of membership in multiple communities. Community Affiliation models with continuous communities include Mixed Membership SBM (MMSBM) (Airoldi et al., 2008) which is an SBM which allows nodes to have mixed memberships in multiple communities, and BigClam (Yang & Leskovec, 2013), which is a Bernouli Poisson model model that scales efficiently for sparse graphs.

It is also worth mentioning the NOCD model presented in (Shchur & Günnemann, 2019) which uses the Bernouli Poisson loss, but embeds the nodes of the graph into the code space using a learned GNN. Another related paper is (Sun et al., 2019),

which models the features and communities separately, learning latent features from which it estimates the community affiliation.

While community affiliations are typically non-negative, there are models where affiliations can be negative. An example is the Signed Stochastic Block Model (SSBM) (Jiang, 2015).

### B.6. Anomaly Detection

Anomaly detection in graphs aims to identify nodes or subgraphs that deviate significantly from the typical patterns within the graph. This is useful in various applications such as network security, fraud detection, and social network analysis. In the unsupervised formulation of the problem, there is no labeled data in the training process, We highlight five works in this direction. GAAN (Chen et al., 2020b) and AEGIS (Ding et al., 2021) use a generative adversarial approach, training a discriminator to distinguish real and fake nodes. Dominant (Ding et al., 2019a) and AnomalyDAE (Fan et al., 2020) identify anomalies via reconstruction errors of a graph autoencoder.

Another work is (Qiao et al., 2024), which considers a semi-supervised setting. Here, there is a relatively small number of available labeled normal nodes during training (nodes that are known to not be anomalies), and the goal is to predict the anomalies in the unknown nodes.

### B.7. Link Prediction

Link prediction in graphs aims to predict missing or future connections between nodes based on the existing structure and node features. This task is critical in various domains such as social networks, biological networks, and recommendation systems. In the supervised setting, a subset of dyads is withheld as a test set, with their labeles (edges or non-edges) omitted. The goal is to predict the connectivity of these omitted dyads using the remaining data, which may also include node features.

We highlight five notable works in this area. The Adamic-Adar index (Adamic & Adar, 2003) henceforth referred to as AA, is a second-order heuristic that assumes shared neighbors with lower degrees contribute more to the likelihood of a link, while high-degree neighbors are less significant. VGAE (Kipf & Welling, 2016) is a variational graph autoencoder, discussed in Section B.3. LINKX (Lim et al., 2021a) decouples the feature and structure representations in a graph and uses a simple framework to process this information separately. GAT (Velickovic et al., 2017) applies an attention mechanism in the message passing process, allowing the model to weight neighboring nodes based on their importance. Another recent work, DisenLink (Zhou et al., 2022), learns disentangled representations of nodes by capturing multiple latent factors, allowing it to model various aspects of node connectivity for link prediction. All models mentioned utilize node features accept for AA.

### B.8. Normalizing Flows

Normalizing flows are deep learning algorithms that estimate probability distributions, and allow an efficient sampling from this distribution. They do so by constructing an invertible coordinate transformation between the unknown target probability space and a standard probability space with a well known distribution (e.g. Gaussian). This transformation is modeled as a deep neural network, composed of a series of basic transformations called *flows*.

**Notations.** Let $\mathcal{F}$ represent the space of the target data, with an unknown probability density function $p_{\mathcal{F}} : \mathcal{F} \to [0, \infty)$. We denote the elements of $\mathcal{F}$ by $\mathbf{f}$.

Let $\mathcal{Z}$ represent a latent space with a known probability density funcion $p_{\mathcal{Z}} : \mathcal{Z} \to [0, \infty)$. We denote the elements of $\mathcal{Z}$ by $\mathbf{z}$. The density function $p_{\mathcal{Z}}$ is often chosen to be a standard isotropic Gaussian with zero mean and identity covariance, denoted by $\mathcal{N}(0, \mathbf{I})$.

**Goal.** The goal in normalizing flows is to learn an invertible transformation $T_{\theta} : \mathcal{F} \to \mathcal{Z}$ (parameterized by $\theta$) which maps the target space $\mathcal{F}$ to the latent space $\mathcal{Z}$, and preserves probabilities. Namely, $T_{\theta}$ should satisfy: for every measurable subset $F \subset \mathcal{F}$ we have

$$\int_F T_{\theta}(\mathbf{f})p_{\mathcal{F}}(\mathbf{f})d\mathbf{f} = \int_{T_{\theta}(F)} p_{\mathcal{Z}}(\mathbf{z})d\mathbf{z}.$$

Here $T_{\theta}(F) = \{z \in \mathcal{Z} \mid \exists \mathbf{f} \in F \text{ such that } T_{\theta}(\mathbf{f}) = \mathbf{z}\}$. Such a transformation can be seen as a change of variable.

**Density Transformation.**  Given an invertible transformation $T_\theta$, the target density $p_\mathcal{F}(\mathbf{f})$ can be expressed in terms of the latent density $p_\mathcal{Z}(\mathbf{z})$. By upholding the constraint that either probability density function has integral 1 over their respective spaces, one can deduce the change of variable formula

$$p_\mathcal{F}(\mathbf{f}) = p_\mathcal{Z}(T_\theta(\mathbf{f})) \left| \det \left( \frac{\partial T_\theta(\mathbf{f})}{\partial \mathbf{f}} \right) \right|.$$

Here, $\frac{\partial T_\theta(\mathbf{f})}{\partial \mathbf{f}}$ denotes the Jacobian matrix of the transformation $T_\theta$ with respect to $\mathbf{F}$ and $\det(\cdot)$ denotes its determinant.

**Sequential Composition of Flows.**  In practice, the transformation $T_\theta$ is modeled as a composition of $L \in \mathbb{N}$ simpler invertible transformations $\{T_{\theta^i}\}_{i=1}^L$ between the consecutive spaces $\{\mathcal{Z}^i\}$ where $\mathcal{Z}^L = \mathcal{F}$ and $\mathcal{Z}^1 = \mathcal{Z}$ in the previous notations. The invertible mappings $T_{\theta^i} : \mathcal{Z}^i \to \mathcal{Z}^{i-1}$ are called flows, and we have

$$T_\theta = T_{\theta^L} \circ T_{\theta^{L-1}} \circ \cdots \circ T_{\theta^1}$$

where $\circ$ denotes function composition. The overall density function for $\mathbf{f}$ then becomes

$$p_\mathcal{F}(\mathbf{f}) = p_\mathcal{Z}(T_\theta(\mathbf{f})) \prod_{i=1}^L \left| \det \left( \frac{\partial T_\theta^i(\mathbf{f}_i)}{\partial \mathbf{f}_i} \right) \right|.$$

Here, $\mathbf{f}_i$ is the intermediate representation after applying the first $i - 1$ flows. The algorithm is optimized by maximum log likelihood, namely, by maximizing

$$\log \left( p_\mathcal{F}(\mathbf{f}) \right)$$
$$= \log \left( p_\mathcal{Z}(T_\theta(\mathbf{f})) \right) + \sum_{i=1}^L \log \left( \left| \det \left( \frac{\partial T_\theta^i(\mathbf{f}_i)}{\partial \mathbf{f}_i} \right) \right| \right),$$

using gradient descent.

Since $T_\theta$ and every $T_\theta^i$ are invertible, information can also flow in the opposite direction in order to sample data. First, a point $\mathbf{z}$ is sampled $\mathbf{z} \sim p_\mathcal{Z}$ and the inverse transformation $T_\theta^{-1}$ is applied to map the sample $\mathbf{z}$ into the data space $\mathcal{F}$. Since $T$ is composed of a series of flows, the generated point in the data space is

$$\mathbf{f} = T_\theta^{-1}(\mathbf{z}) = T_{\theta^1}^{-1} \circ T_{\theta^2}^{-1} \circ \cdots \circ T_{\theta^L}^{-1}(\mathbf{z}).$$

**RealNVP.**  One popular method of implementing normalizing flow is the *Real Valued Non-Volume Preserving (RealNVP)*. In this model, the flows $T_{\theta^i}$ are modeled as mappings called *coupling blocks*. In a coupling block, the input vector $\mathbf{f}^i$ (or $\mathbf{z}^i$, in the generative direction) is split into two vectors: $\mathbf{f}_A^i$ and $\mathbf{f}_B^i$, each the same dimension. Namely, $\mathbf{f}^i = (\mathbf{f}_A^i, \mathbf{f}_B^i)$. The flow $T_{\theta^i}$ is then defined to be

$$\mathbf{f}_A^{i-1} = \mathbf{f}_A^i \tag{22}$$
$$\mathbf{f}_B^{i-1} = \mathbf{f}_B^i \odot \mathbf{s}_{\theta^i}(\mathbf{f}_A^i) + \mathbf{t}_{\theta^i}(\mathbf{f}_A^i) \tag{23}$$

Where $(\mathbf{s}_{\theta^i}(\cdot), \mathbf{t}_{\theta^i}(\cdot))$ are Multi Layer Perceptrons (MLPs) parameterized by $\theta^i$, and $\odot$ is elementwise product. In addition, the elements of $\mathbf{f}^i$ are permuted at every $i$ before applying the coupling block, using predefined permutations, that are parts of the hyperparameters of the model. It is easy to see that the Jacobian $\left( \frac{\partial T_\theta^i(\mathbf{f}_i)}{\partial \mathbf{f}_i} \right)$ is a diagonal matrix with 1 for the first $\dim(\mathbf{f}_A^i)$ elements, and $s_{\theta^i}(\mathbf{f}_A^i)$ for the last $\dim(\mathbf{f}_B^i)$ elements. This makes the log of the determinant of the Jacobian

$$\log \left( \left| \det \left( \frac{\partial T_\theta^i(\mathbf{f}_i)}{\partial \mathbf{f}_i} \right) \right| \right) = \sum_{j=0}^{\dim(\mathbf{f}_B^i)} \log \left( \mathbf{s}_{\theta^i}(\mathbf{f}_A^i)_j \right). \tag{24}$$

Hence, this determinant is easily computed in practice.

For generation, the inverse transformation can be calculated easily as

$$\mathbf{z}_A^{i+1} = \mathbf{z}_A^i$$

$$\mathbf{z}_B^{i+1} = \frac{\mathbf{z}_B^i - \mathbf{t}(\mathbf{z}_A^i)}{\mathbf{s}(\mathbf{z}_A^i)}$$

where the division is elementwise. Here, the log determinant of the Jacobian can be calculated similarly to (24), applied to $1/\mathbf{s}(\mathbf{z}_A^i)$.

### B.9. The Lorentz Inner Product

The Lorentz inner product is a bilinear form used in the context of special relativity to describe the spacetime structure. In a four-dimensional spacetime, the Lorentz inner product between two vectors $\mathbf{v}$ and $\mathbf{w}$ is given by:

$$\langle \mathbf{v}, \mathbf{w} \rangle = v_0 w_0 - v_1 w_1 - v_2 w_2 - v_3 w_3$$

Here, $v_0$ and $w_0$ represent the "time" components, while $v_1, v_2, v_3$ and $w_1, w_2, w_3$ represent the "spatial" components. The negative sign in front of the time component is what distinguishes the Lorentz inner product from the standard Euclidean inner product, making it suitable for modeling the geometry of spacetime where time and space are treated differently. Note that due to the subtraction, the Lorenz inner product is in fact not an inner product as it is not positive definite. In the context of special relativity, the points for which the Lorenz product remain positive define the so called "light cone" structure of spacetime, separating events into those that are causally connected and those that are not.

The Lorentz inner product is a specific example of a broader class of inner products known as **pseudo-Euclidean inner products**. In a pseudo-Euclidean space, the inner product can have a mixture of positive and negative signs, leading to different geometric properties. These spaces generalize the concept of Euclidean space by allowing for non-positive definite metrics.

## C. PClam and PieClam as Graphons

A graphon is a model which can be seen as a graph generative model that extends SBMs. A graphon (Borgs et al., 2007; Lovász, 2012) can be seen as a weighted graph with a "continuous" node set $[0, 1]$.

**Definition C.1.** The space of graphons $\mathcal{W}$ is defined to be the set of all measurable function $W : [0, 1]^2 \to [0, 1]$ which are symmetric, namely $W(x, y) = W(y, x)$.

The edge weight $W(x, y)$ of a graphon $W \in \mathcal{W}$ can be seen as the probability of having an edge between node $x$ and node $y$. Given a graphon $W$, a random graph is generated by sampling independent uniform nodes $\{X_n\}$ from the graphon domain $[0, 1]$, and connecting each pair $X_n, X_m$ in probability $W(X_n, X_m)$ to obtain the edges of the graph.

We next show that Clam models with prior are special cases of graphon models. Note that the prior $p$ defines a standard atomless probability spaces over the community affiliation space $\mathbb{R}^K$. Since all standard atomeless probability spaces are equivalent, there is a probability preserving a.e. bijection $\xi_p : [0, 1] \to \mathbb{R}^K$ that maps the prior probability to the uniform probability over $[0, 1]$. Now, the PieClam model for generating a graph of $N$ nodes can be written as follows.

- Sample $N$ points $\{X_n\}_{n=1}^N \subset [0, 1]$ uniformly. Observe that $\{\xi_p(X_n)\}_{n=1}^N \subset \mathbb{R}^K$ are indepndent samples via the probability density $p$.

- Connect the points according to the BigClam of IeClam models $P(n \sim m | \xi_p(X_n), \xi_p(X_m))$ (1) or (7).

This shows that PClam and PieClam coincide with the generative graphon model $W(x, y) = P(n \sim m | \xi_p(X), \xi_p(Y))$ where $P$ is defined either by (1) or by (7).

## D. Extended Details on Experiments

### D.1. Additional Architecture Details in PieClam and PClam

**Added Affiliation Noise.** When training the prior, overfitting may cause the probability to spike around certain areas in the affiliation space, e.g., around the affiliation features of the nodes. To avoid this issue, we add gaussian noise to the

| Method | Reddit | Elliptic | Photo |
|---|---|---|---|
| (S)- IeClam | **64.12 ± 0.25** | 43.58 ± 0.15 | *57.67 ± 1.79 |
| (S) - PieClam | *63.97 ± 0.50 | 43.47 ± 00.12 | 58.98 ± 2.51 |
| (P) - PieClam | 46.76 ± 2.50 | **63.18 ± 3.02** | 45.73 ± 7.03 |
| (PS) - PieClam | 63.99 ± 0.50 | 53.81 ± 1.63 | **58.99 ± 2.52** |
| DOMINANT | 51.1 | 29.6 | 51.4 |
| AnomalyDAE | 50.9 | *49.6 | 50.7 |
| OCGNN | 52.5 | 25.8 | 53.1 |
| AEGIS | 53.5 | 45.5 | 55.2 |
| GAAN | 52.2 | 25.9 | 43.0 |
| TAM | 60.6 | 40.4 | 56.8 |

Table 4: Comparison of Clam anomaly detectors with competing methods. First place in **boldface**, second with underline, third with *star. We observe that our methods are first place on all datasets. Moreover, S- IeClam, PS-PieClam and S BigClam each beats the competing methods in two out of the three datasets . The accuracy metric is areas under curve (AUC).

| Method | Reddit | Elliptic | Photo |
|---|---|---|---|
| (S)- IeClam | **64.12 ± 0.25** | 43.58 ± 0.15 | *57.67 ± 1.79 |
| (S) - PieClam | *63.97 ± 0.50 | 43.47 ± 0.12 | 58.98 ± 2.51 |
| (P) - PieClam | 46.76 ± 2.50 | **63.18 ± 3.02** | 45.73 ± 7.03 |
| (PS) - PieClam | 63.99 ± 0.50 | 53.81 ± 1.63 | **58.99 ± 2.52** |
| (S) - PieClam (No Attr) | 63.46 ± 0.15 | 43.50 ± 0.13 | 58.11 ± 2.73 |
| (P) - PieClam (No Attr) | 45.30 ± 1.42 | *50.25 ± 0.8 | 48.67 ± 10.47 |
| (PS) - PieClam (No Attr) | 63.51 ± 0.15 | 43.50 ± 0.13 | 58.12 ± 2.75 |

Table 5: Comparison of attributed and unattributed PieClam anomaly detection.

affiliation vector as a regularization at each step of training the normalizing flow prior model. The normalizing flow model then transforms the same point to a slightly different location in the code space. The noise optimization therefore provides a resolution to the prior, smoothing the distribution in the affiliation space. Noise addition is the primary regularization method we have used when training the prior in all of our experiments.

**Densification.** For very sparse datasets, Clam models may not behave well. In such cases, the community structure is sometimes unstable, where the same node can find itself in different communities based on slightly different initial conditions. In order to strengthen the connections within communities, we apply a two-hop densification scheme on very sparse graphs. Namely, we connect two disjoint nodes $n, m$ with an edge if there is a third node $k$ for which $n \sim k$ and $m \sim k$. We find that this scheme improves anomaly detection on Elliptic, Photo and Reddit datasets. Densification may strengthen the community structure in some datasets, but can also destroy it in others. In the experiments on synthetic datasets we did not use densification, as these graphs are relatively dense.

### D.2. Anomaly detection

**Metric.** In order to measure the accuracy of the anomaly detection classification we use the area under the Receiver Operating Characteristic (ROC) curve (Peterson et al., 1954). The curve is a plot of the true positive rate (TPR) against the false positive rate (FPR) for the range of the threshold values between the value that classifies all samples as true and the value that classifies all as false. The area under the curve signifies the general tendency of the curve toward the point (0,1) for which FPR= 0 and TPR= 1.

**Experimental Details.** For the anomaly detection results, each model is assigned a single configuration that is used across all datasets. Every configuration is evaluated by training the model 10 times per dataset, and the results are reported as mean and standard deviation. The detailed results with error bars are shown in Table 5. To select the global hyperparameters, we first conducted a wide scan over parameter ranges, followed by a finer search around the most promising regions.

| Method | Squirrel | Photo | Texas | John's Hopkins 55 |
|---|---|---|---|---|
| PieClam | **98.7 ±0.0** | **98.4 ±0.0** | **85.0 ± 2.0** | 95.5 ±0.0* |
| BigClam | 98.5 ± 0.0 | 97.4 ± 0.1* | 78.2 ± 3.0* | 94.9 ± 0.0 |
| VGAE | 98.2 ±0.1 | 94.9 ±0.8 | 68.6 ±4.2 | 92.8 ±0.2 |
| GAT | 98.0 ±0.0 | 97.3 ±0.3 | 68.5 ±5.4 | 94.3 ±0.5 |
| LINKX | 98.1 ±0.3 | 97.0 ±0.2 | 75.8 ±4.7 | 93.4 ±0.3 |
| AA | 97.1 ±0.4 | 97.4 ±0.4 | 53.1 ±6.2 | 96.1 ±0.5 |
| DisenLink | 98.3 ±0.1* | 97.9 ±0.1 | 81.0 ± 4.0 | **97.5 ±0.1** |

Table 6: Comparison of Clam link predictors with competing models. First place in **boldface**, second with underline, third with *star

Furthermore, in this section we also run our anomaly detection methods without using the node features, namely, only using the graph structure. We find that we comparable results on Photo and Reddit with or without node features, but that the detection on Elliptic decreases significantly. Still, anomaly detection on Elliptic without node features is competitive with state of the art models that do use node features. The results are in Table 5.

We proceed to compare the optimization that includes the node features to the optimization that didn't. The comparison is presented in

**T-Test Comparison Summary.**    We conducted t-tests to determine if node features/attributes ("With Features") significantly improve performance compared to not using attributes ("Without Features"), across the Reddit, Elliptic, and Photo datasets for each method (S, P, PS). Even though S method is not affected directly by the affiliation vectors, the latter can affect the convergence of the prior in the affiliation space and have an indirect effect.

- **Reddit Dataset**: Attributes did not significantly improve performance for the "Star" (S) and "Prior Star" (PS) methods. However, "Prior" (P) did show a significant improvement with attributes (p-value=0.0002).

- **Elliptic Dataset**: "Prior" (P) and "Prior Star" (PS) both showed significant improvements with attributes (p-value<0.0001). However, the "Star" (S) method did not significantly benefit from attributes.

- **Photo Dataset**: The "Star" (S) method did not show significant difference (p-value=0.6864). The "Prior Star" (PS) method, however, did show a significant benefit from attributes (p-value<0.0001), and "Prior" (P) had a borderline significant result (p-value=0.0529).

In conclusion, attributes did not make a difference for the "Star" (S) method across all datasets (as would be expected), and for the "Prior Star" (PS) method in the Reddit dataset. However, the "Prior" (P) and "Prior Star" (PS) methods mostly showed improvement in the Elliptic dataset.

### D.3. Link Prediction

**Probabilistic Model with Omitted Dyads.**    When doing link prediction (as explained in Section B.7) we remove a portion of all dyads from computation during training, treating the omitted dyads neither as edges nor non-edges. Having non-affiliated dyads modifies the PieClam log likelihood as explained next.

Recall that $E$ denotes the set of all edges in the graph. Denote $D = \{(n, n) | n \in [N]\}$. Denote by $\bar{E} = [N]^2 \setminus (E \cup D)$ the set of non-edges. Denote the sets of omitted edges and non-edges by $\check{E}$ and $\check{\bar{E}}$ respectively. We assume that the number of omitted dyads $\check{E} \dot{\cup} \check{\bar{E}}$ is of the same order as the number of edges $E$. Hence, for sparse graphs $|\check{E} \dot{\cup} \check{\bar{E}}| \ll N^2$. The PieClam loss which ignores the omitted dyads now is defined to be

$$2\hat{l}(\mathbf{F}, E, \check{E}, \check{\bar{E}}) = 2 \sum_{n \in [N]} \log(p(\mathbf{f}_n)) + \sum_{(n,m) \in E \setminus \check{E}} \log(1 - e^{-\mathbf{f}_n^\top \mathbf{L} \mathbf{f}_m}) - \sum_{(n,m) \in \bar{E} \setminus \check{\bar{E}}} \mathbf{f}_n^\top \mathbf{L} \mathbf{f}_m \qquad (25)$$

Next we formulate the PieClam loss as an efficient "message passing," requiring only $O(|E|)$ operations. First, note that

$$[N]^2 \setminus (\bar{E} \setminus \check{\bar{E}}) = E \dot{\cup} \check{\bar{E}} \dot{\cup} D.$$

Therefore,

$$\sum_{(n,m)\in E\setminus \check{E}} \log(1 - e^{-\mathbf{f}_n^\top \mathbf{L}\mathbf{f}_m}) - \sum_{(n,m)\in \bar{E}\setminus \check{E}} \mathbf{f}_n^\top \mathbf{L}\mathbf{f}_m$$

$$= \sum_{(n,m)\in E\setminus \check{E}} \log(1 - e^{-\mathbf{f}_n^\top \mathbf{L}\mathbf{f}_m}) - \sum_{(n,m)\in [N]^2} \mathbf{f}_n^\top \mathbf{L}\mathbf{f}_m + \sum_{(n,m)\in [N]^2\setminus(\bar{E}\setminus \check{E})} \mathbf{f}_n^\top \mathbf{L}\mathbf{f}_m$$

$$= \sum_{(n,m)\in E\setminus \check{E}} \log(1 - e^{-\mathbf{f}_n^\top \mathbf{L}\mathbf{f}_m}) - \sum_{(n,m)\in [N]^2} \mathbf{f}_n^\top \mathbf{L}\mathbf{f}_m + \sum_{(n,m)\in E\dot{\cup}\check{E}\dot{\cup}D} \mathbf{f}_n^\top \mathbf{L}\mathbf{f}_m$$

$$= \sum_{(n,m)\in E\setminus \check{E}} \log(1 - e^{-\mathbf{f}_n^\top \mathbf{L}\mathbf{f}_m}) - \sum_{n\in[N]} \mathbf{f}_n^\top \sum_{m\in[N]} \mathbf{L}\mathbf{f}_m + \sum_{(n,m)\in E\dot{\cup}\check{E}\dot{\cup}D} \mathbf{f}_n^\top \mathbf{L}\mathbf{f}_m.$$

Hence,

$$2\hat{l}(\mathbf{F}, E, \check{E}, \check{\check{E}}) = 2\sum_{n\in[N]} \log(p(\mathbf{f}_n)) + \sum_{(n,m)\in E\setminus \check{E}} \log(1 - e^{-\mathbf{f}_n^\top \mathbf{L}\mathbf{f}_m})$$
$$- \sum_{n\in[N]} \mathbf{f}_n^\top \sum_{m\in[N]} \mathbf{L}\mathbf{f}_m + \sum_{(n,m)\in E\dot{\cup}\check{\check{E}}\dot{\cup}D} \mathbf{f}_n^\top \mathbf{L}\mathbf{f}_m, \tag{26}$$

and gradient ascent with respect to this formulation is an efficient message passing algorithm.

Developing the expression further, we achieve an equivalent form that resembles the original clam loss.

First we decompose the last term into

$$\sum_{(n,m)\in E\dot{\cup}\check{\check{E}}\dot{\cup}D} \mathbf{f}_n^\top \mathbf{L}\mathbf{f}_m = \sum_{(n,m)\in E\setminus \check{E}} \mathbf{f}_n^\top \mathbf{L}\mathbf{f}_m + \sum_{(n,m)\in \check{E}\dot{\cup}\check{\check{E}}\dot{\cup}D} \mathbf{f}_n^\top \mathbf{L}\mathbf{f}_m.$$

We substitute this sum into Equation 26 to get

$$2\hat{l}(\mathbf{F}, E, \check{E}, \check{\check{E}}) = 2\sum_{n\in[N]} \log(p(\mathbf{f}_n)) + \sum_{(n,m)\in E\setminus \check{E}} \log(1 - e^{-\mathbf{f}_n^\top \mathbf{L}\mathbf{f}_m})$$
$$- \sum_{n\in[N]} \mathbf{f}_n^\top \sum_{m\in[N]} \mathbf{L}\mathbf{f}_m + \sum_{(n,m)\in E\setminus \check{E}} \log(e^{\mathbf{f}_n^\top \mathbf{L}\mathbf{f}_m}) + \sum_{(n,m)\in \check{E}\dot{\cup}\check{\check{E}}\dot{\cup}D} \mathbf{f}_n^\top \mathbf{L}\mathbf{f}_m.$$

We unify the second and fourth sums and use logarithm identities to get

$$2\hat{l}(\mathbf{F}, E, \check{E}, \check{\check{E}}) = \left(2\sum_{n\in[N]} \log(p(\mathbf{f}_n)) + \sum_{(n,m)\in E\setminus \check{E}} \log(e^{\mathbf{f}_n^\top \mathbf{L}\mathbf{f}_m} - 1) - \sum_{n\in[N]} \mathbf{f}_n^\top \sum_{m\in[N]} \mathbf{L}\mathbf{f}_m + \sum_{(n)\in[N]} \mathbf{f}_n^\top \mathbf{L}\mathbf{f}_n\right) + \sum_{(n,m)\in \check{E}\dot{\cup}\check{\check{E}}} \mathbf{f}_n^\top \mathbf{L}\mathbf{f}_m. \tag{27}$$

The first three terms (in brackets) of 27 make the original clam loss (Equation 10) on the retained set of edges $E \setminus \check{E}$. The forth term represents an MPNN on the graph $([N], \check{E}\dot{\cup}\check{\check{E}})$. Since $\check{E}\dot{\cup}\check{\check{E}}$ is strictly contained in $E\dot{\cup}\check{\check{E}}$ we use Equation 27 for the link prediction implementation.

**Experimental Setup** In the experiments, we evaluate the model's ability to classify edges and non-edges by constructing a ROC curve as described in Section B.6.

Note that, since the prior is not used in classification, there is no added benefit from utilizing node features when training the prior, as proposed in 2.4. Therefore, the link prediction algorithm relies solely on the graph's topological structure and the prior functions only as regularization. We use the scheme from (Zhou et al., 2022), where for each dataset, we generate 10 randomly sampled test sets, each containing 10% of the edges, along with 5 randomly sampled non-edges for every omitted edge. For each split, we perform hyperparameter selection using a separate 5% validation set, also paired with 5 non-edges per edge. We report the mean accuracy across the 10 test sets, along with the standard error of the mean. To select the hyperparameters for each split, we first conducted a wide scan over parameter ranges, followed by a finer search around the most promising regions. To ease reproducibility, we use a single configuration per model and dataset across all sampled test sets. The test sets are available in our GitHub repository.

| | Model | Dim. | Feat. Iter. | Prior Iter. | Prior Step | S Reg. | First Optimization |
|---|---|---|---|---|---|---|---|
| Squirrel | PieClam | 90 | 700 | 2000 | 5e-6 | 0.1 | Feature |
| Photo | PClam | 150 | 700 | 1300 | 3e-6 | - | Feature |
| Texas | PieClam | 22 | 2000 | 1500 | 5e-6 | 0.0 | Prior |
| JH55 | PClam | 200 | 750 | 1800 | 3e-6 | - | Feature |

Table 7: Link prediction hyperparameters for each of the datasets. Full configuration details are available in the project's GitHub repository.

**Datasets and Baselines.**   We conduct these experiments on four datasets: Facebook100's Johns Hopkins 55 (Lim et al., 2021b), Amazon Photo (Shchur et al., 2018b), WebKB's Texas and Wikipedia's Squirrel (Pei et al., 2020). We compare ourselves to the baseline methods AA (Adamic & Adar, 2003), VGAE (Kipf & Welling, 2016), GAT (Velickovic et al., 2017), LINKX (Lim et al., 2021a), DisenLink (Zhou et al., 2022) and BigClam.

**Hyperparameter Configuration.**   As was mentioned in D.3, hyperparameters are selected separately for each dataset and model by randomly selecting a validation set outside of the test set. Since all four Clam models are a special case of PieClam, we present only the best performing model and treat it as a hyperparameter. We describe the hyperparameter configuration for each of the models in Table 7. The columns of the table signify the Clam model, AS dimension, number of feature iterations, feature step size, number of prior weight iterations, the prior weight step size, the added noise amplitude, s community regularization and the first function in the alternation. The following hyperparameters are constant across all datasets: feature step size (5e-6), $l_1$ regularization (1), Noise amplitude (0.1) the number of alternations (7), scheduler step size (3) and the scheduler step size (0.5).

**Results**   The results on the datasets and the comparison to the other baseline models is presented in Table 6. Baseline results are taken from Table 2 in (Zhou et al., 2022).

## D.4. Reconstructing Synthetic Priors

We consider two ground-truth synthetic priors $p_{big} : \mathbb{R}_+^2 \to [0, \infty)$ in PClam, and $p_{ie} : \mathcal{T}^1 \to [0, \infty)$ in PieClam. We sample $N = 500$ points from each of the priors, decode the corresponding PClam and PieClam graphs, and sample simple graphs from the random Bernoulli edges as shown in figure 4. Then, given these sampled graphs, we reset their affiliation features to random vectors and fit a BigClam and a PClam to the graph sampled from $p_{big}$, and an IeClam and a PieClam to the graph that was sampled from $p_{ie}$.

The optimization results are shown in Figures 5 and 6. Both the priored and unpriored models effectively learned the feature reconstructions in the feature spaces $\mathcal{T}$ and $\mathbb{R}_+^2$. Notably, the features in the priored models are more localized, supporting the claim that the prior acts as a regularizer.

The priors themselves are well-approximated, as illustrated in Figure 6, with the quality of approximation improving as more nodes are sampled from the priors.

## D.5. SBM Reconstruction And Distance Convergence

We extend the experiment in Section 4.2 by considering another synthetic SBM, which is off-diagonal dominant. As in Section 4.2, we sample a simple graph with $N = 210$ nodes from the SBM. The non zero probability blocks have a probability of 0.5. We estimate the log cut distance between the Clam models (BigClam and IeClam) and the SBM during training. We consider for both methods affiliation spaces of dimensions 2, 4 and 6. In addition, we calculate the cut distance and l2 errors between the Clam models and the SBM. The results are shown in Figures 8–13.

## D.6. Ablation Study

**$l_1$ and s Regularization.**   The $l_1$ regularization is adapted from the BigClam original architecture and shows improvement when optimizing the models. This regularization type adds a constant update in all of the directions, which means that for every node $n$ and it's affiliation feature $\mathbf{f}_n$, components that are not updated are moved towards 0 regardless of their magnitude. This constant push towards 0 forces sparsity in the affiliation matrix and also acts as a regularization. The s regularization is an $l_1$ regularization of the s features and has shown effective for the Squirrel dataset.

**Noise Addition.**   Due to the high expressivity of the normalizing flow model, the neural network tends to learn a narrow distribution around the affiliation features. This overfitting tendency cannot be alleviated by simply reducing the network parameters without significantly compromising the expressivity of the prior. Noise addition offers a way to control the prior's resolution while preserving its ability to learn complex distributions. An example comparing optimization results with and without noise addition is shown in Figure 7.

**Densification.**   The densification method described in Section D.1 can yield varying results depending on the internal structure of the graph. To illustrate the impact of densification, we conducted 10 anomaly detection experiments, following the setup from Section 4.3,

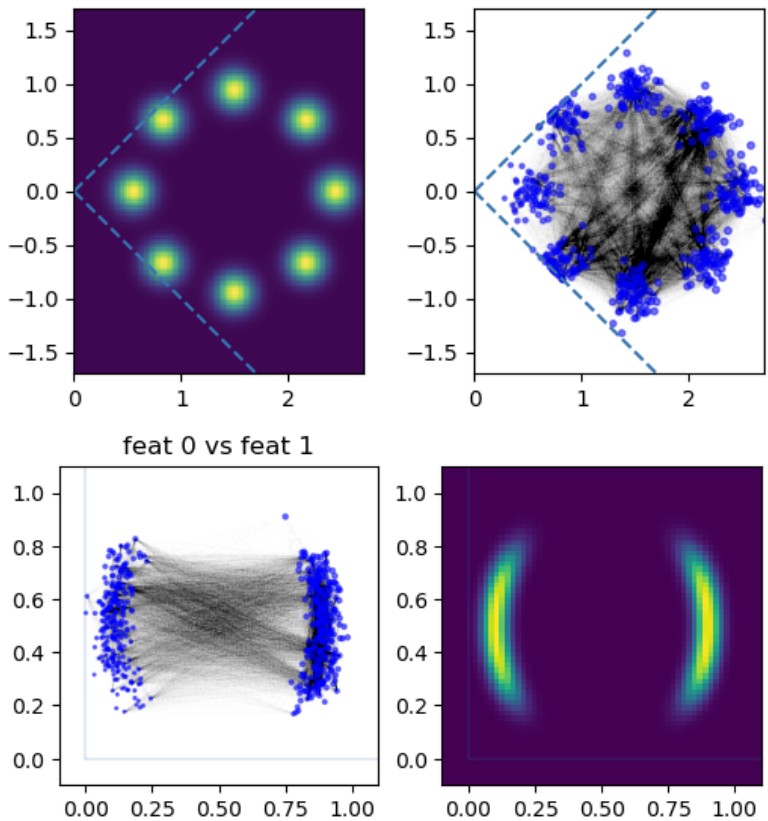

Figure 4: Right to left: ground truth synthetic prior and sampled graph. Top: graph sampled in $\mathcal{T}$ using PieClam. Bottom: graph sampled in $\mathbb{R}_+^2$ using Pclam.

but without applying densification. The results are shown in Table 8.

| Method | Reddit | Elliptic | Photo |
|---|---|---|---|
| (S) - PieClam | **0.6320 ± 0.0054** | **0.4354 ± 0.0005** | **0.5727 ± 0.0151** |
| (S) - PieClam (No Dens.) | 0.5480 ± 0.0022 | 0.4009 ± 0.0021 | 0.5293 ± 0.0046 |
| (P) - PieClam | 0.5089 ± 0.0219 | 0.6201 ± 0.0176 | 0.4252 ± 0.0074 |
| (P) - PieClam (No Dens.) | 0.5088 ± 0.0383 | **0.6378 ± 0.0245** | 0.4325 ± 0.0089 |
| (PS) - PieClam | **0.6329 ± 0.0052** | 0.5294 ± 0.0089 | **0.5289 ± 0.0127** |
| (PS) - PieClam (No Dens,) | 0.5466 ± 0.0047 | **0.6181 ± 0.0233** | 0.4538 ± 0.0030 |

Table 8: Comparison of PieClam on densified and non densified datasets. **Boldface** denotes a significantly better result (difference of over 2 standard deviations), underline denotes an insignificant difference.

As seen in Table 8, densification generally improved results across the datasets, except for the Elliptic dataset using the P and PS methods, where the undensified datasets performed significantly better. For the Reddit and Photo datasets with the P method, the difference was not significant.

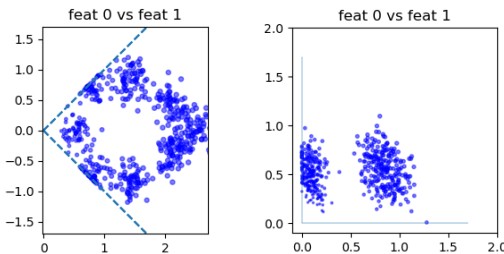

Figure 5: Left to Right: IeClam and BigClam Reconstructions of features sampled from synthetic priors

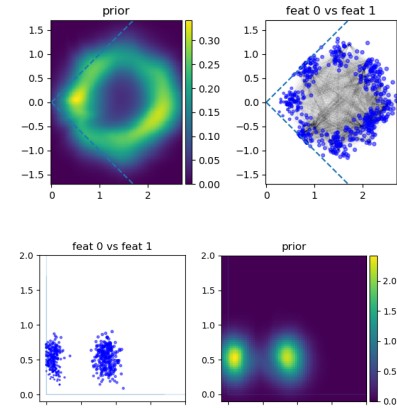

Figure 6: Left to right: reconstructed nodes, reconstructed prior. Top to botton: PieClam in $\mathcal{T}$, PClam in $\mathbb{R}_+^{\not\leq}$

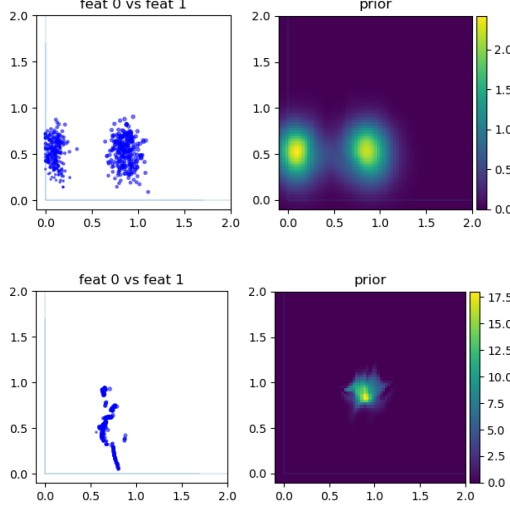

Figure 7: Left to right: reconstructed nodes, reconstructed prior. Top to botton: With and without noise addition.

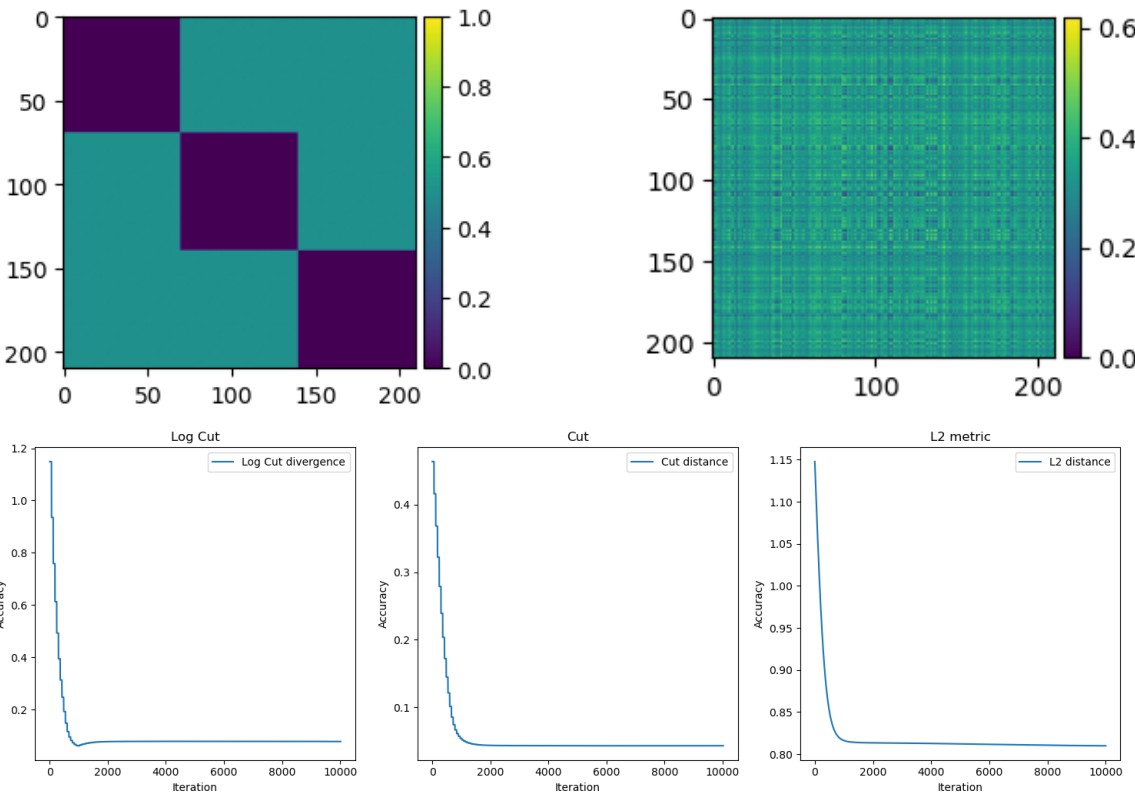

Figure 8: Left to right: Target SBM. Fitted BigClam graph with two communities. Error as a function of optimization iteration where error is, left to right, log cut distance, cut distance, l2 distance. After convergence, the log cut distance between the SBM and BigClam is 0.0776.

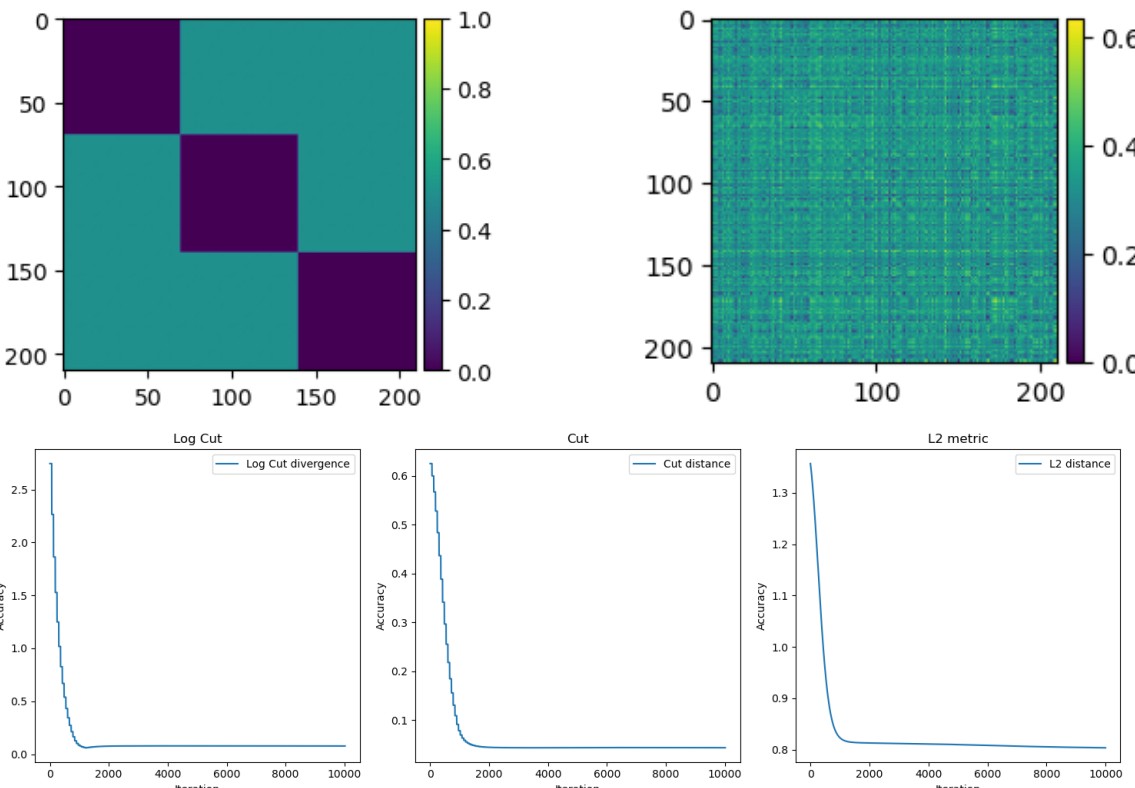

Figure 9: Left to right: Target SBM. Fitted BigClam graph with four communities. Error as a function of optimization iteration where error is, left to right, log cut distance, cut distance, l2 distance. After convergence, the log cut distance between the SBM and BigClam is 0.0775.

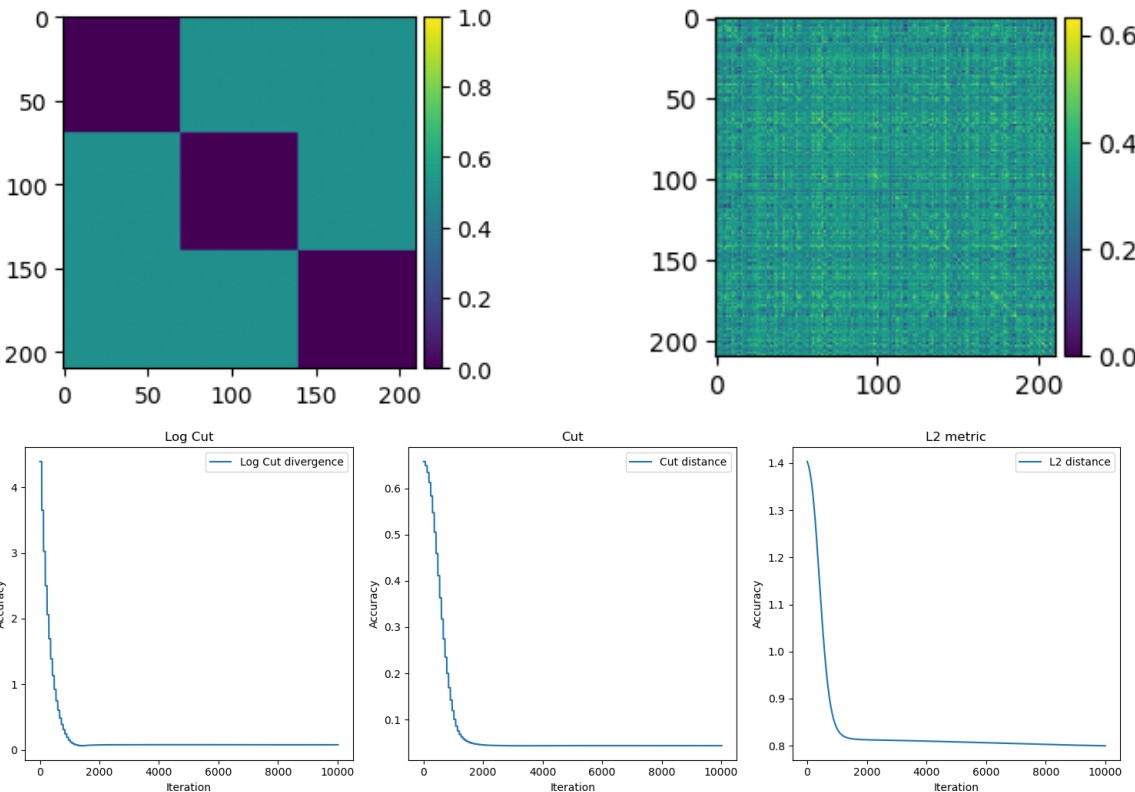

Figure 10: Left to right: Target SBM. Fitted BigClam graph with six communities. Error as a function of optimization iteration where error is, left to right, log cut distance, cut distance, l2 distance. After convergence, the log cut distance between the SBM and BigClam is 0.0776.

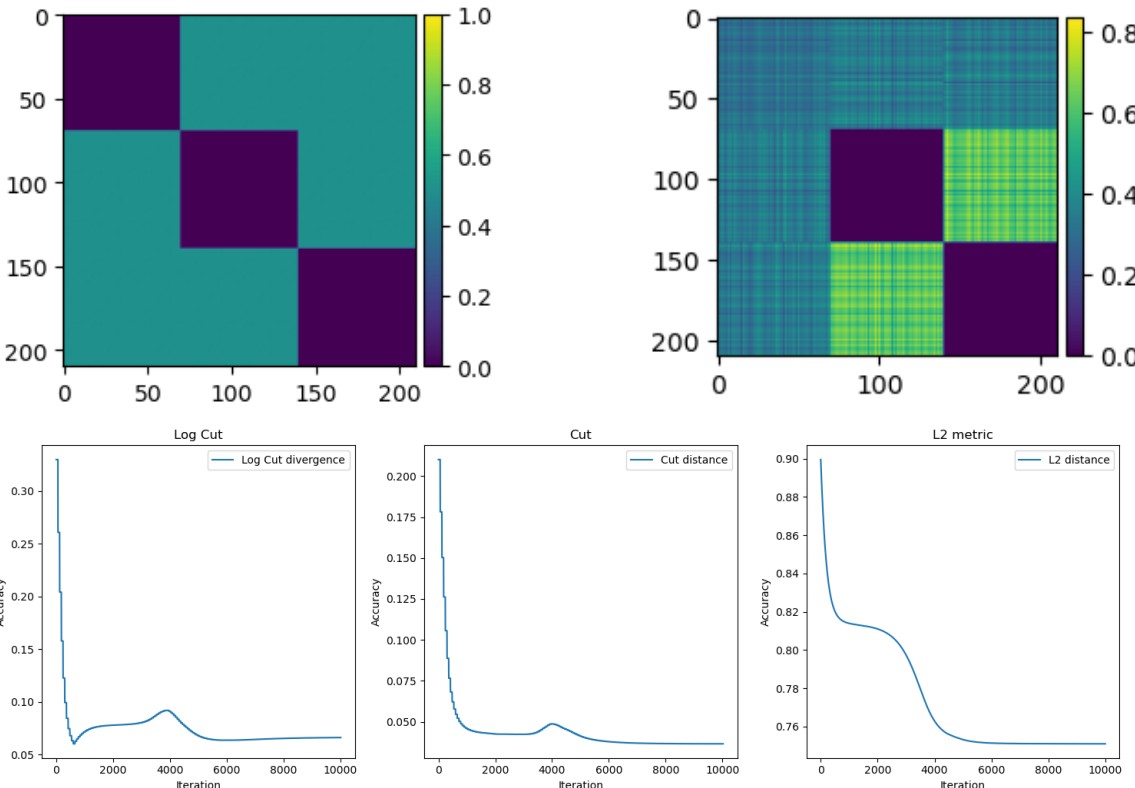

Figure 11: Left to right: Target SBM. Fitted IeClam graph with two communities. Error as a function of optimization iteration where error is, left to right, log cut distance, cut distance, l2 distance. After convergence, the log cut distance between the SBM and IeClam is 0.0662.

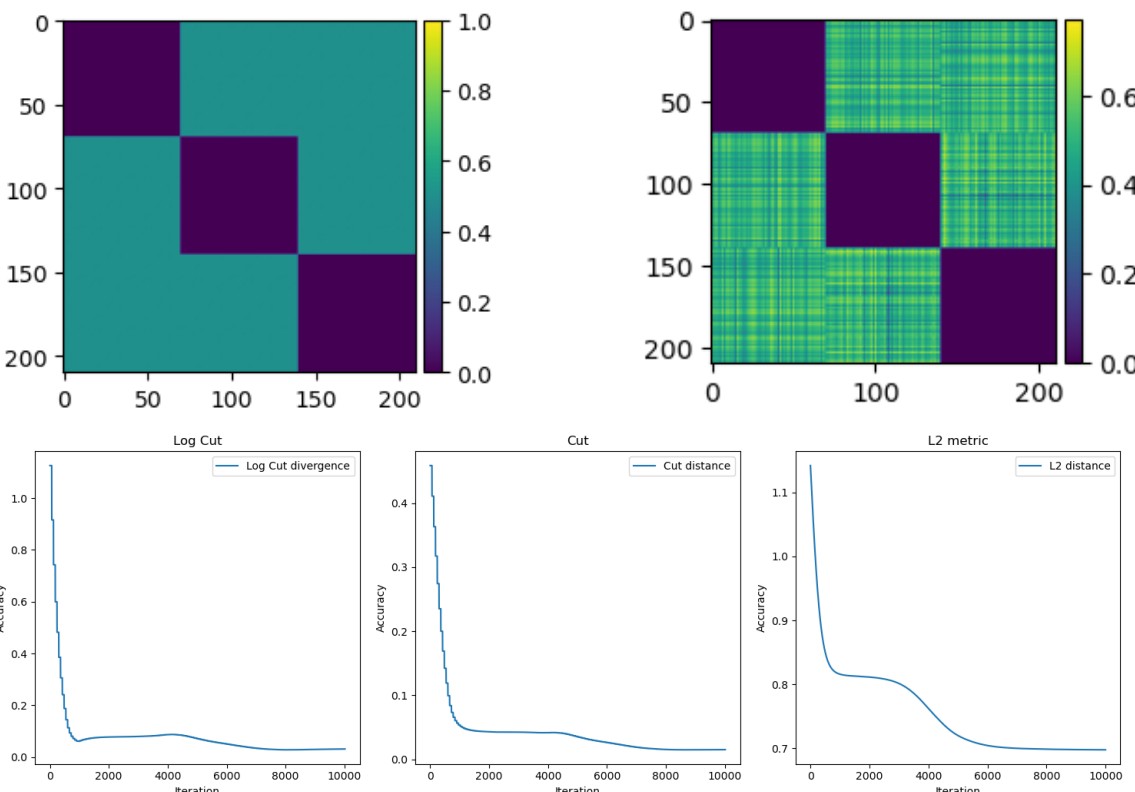

Figure 12: Left to right: Target SBM. Fitted IeClam graph with four communities (two inclusive and two exclusive). Error as a function of optimization iteration where error is, left to right, log cut distance, cut distance, l2 distance. After convergence, the log cut distance between the SBM and IeClam is 0.0312.

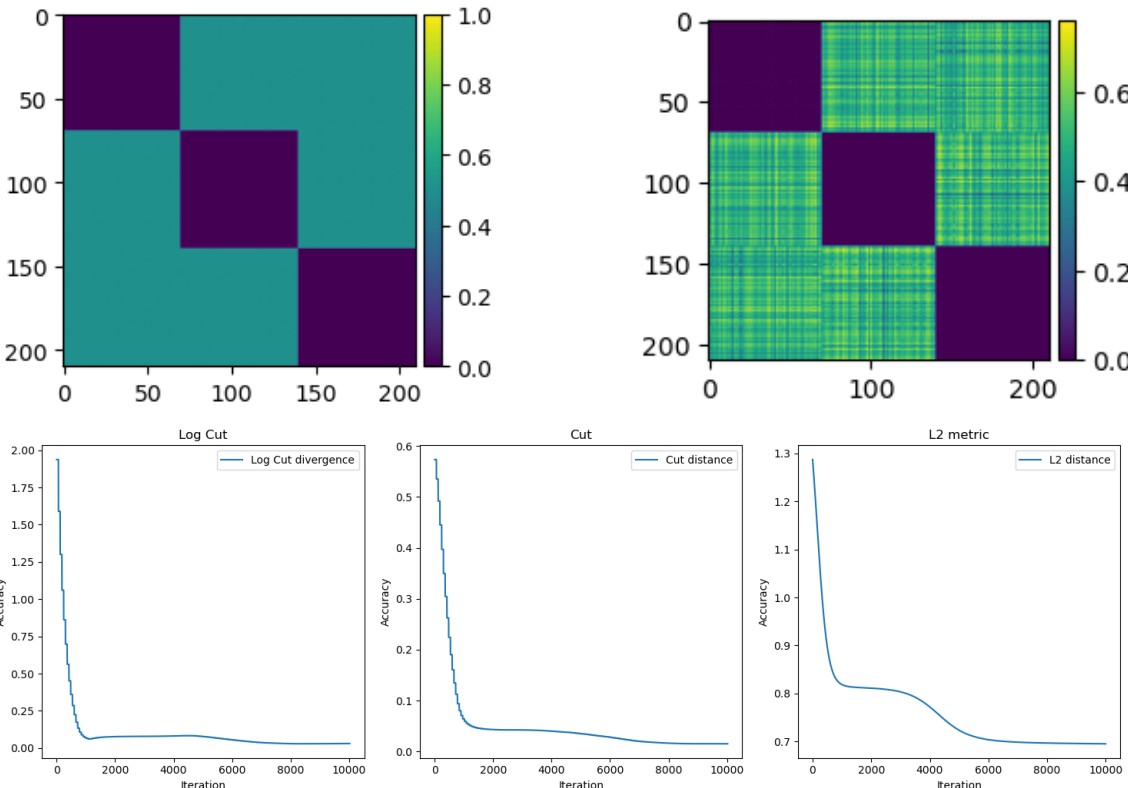

Figure 13: Left to right: Target SBM. Fitted IeClam graph with six communities (three inclusive and three exclusive). Error as a function of optimization iteration where error is, left to right, log cut distance, cut distance, l2 distance. After convergence, the log cut distance between the SBM and IeClam is 0.0309.

