# OpenReview forum: "PieClam: A Universal Graph Autoencoder Based on Overlapping Inclusive and Exclusive Communities"
_ICML.cc/2025/Conference — ICML 2025 poster_

### Official Review · Reviewer_WAjU · 2025-03-15

**Overall Recommendation:** 4

**Summary:**

The document presents PieClam (Prior Inclusive Exclusive Cluster Affiliation Model), a novel graph autoencoder that enhances the existing BigClam framework by utilizing overlapping inclusive and exclusive community structures for node representation learning. PieClam introduces a new log cut distance for measuring graph similarity and demonstrates universality in approximating any graph structure with a fixed number of communities, surpassing prior models in link prediction and anomaly detection tasks. Experiments validate the model's effectiveness across benchmark datasets, showing that it achieves competitive performance while offering a generative capability through prior distribution learning in the community affiliation space.

**Claims And Evidence:**

The claims presented in the manuscript are well-supported by the experiments conducted and the theoretical proofs provided. The authors assert that PieClam can accurately reconstruct graphs and achieve competitive link prediction and anomaly detection outcomes. Evidence for these claims includes:
Comprehensive experimental results on multiple datasets, including Squirrel and Texas, which confirm the model's performance against established baselines such as AA, VGAE, GAT, LINKX, and Disenlink.
The introduction of log cut distance, a new metric aimed at quantifying similarity between graphs, which is substantiated by both empirical and theoretical discussions in the paper.
Theoretical proofs (Theorems 3.7 and 3.8) confirming the universality of the model, indicating its capability to approximate various graph structures efficiently.
While most claims are convincingly backed by data, further clarity on the implications of using the log cut distance in sparse graphs would enhance the overall argument.

**Essential References Not Discussed:**

The manuscript could benefit from including related works that have addressed the integration of features in graph models or explored universal properties of graph representations. For instance:
Recent advancements in normalizing flows for better feature representation could establish a clearer context for PieClam's approach.
Works exploring the strengths and weaknesses of using similar graph distance measures in sparse networks should also be cited.
Inclusion of these references would provide a more rounded understanding of how PieClam positions itself within contemporary research on graph models.

**Experimental Designs Or Analyses:**

The experimental framework and analyses appear sound, demonstrating the validity of the proposed model. Key considerations include:
The selection of datasets (such as Squirrel and Texas) is relevant and conducive to assessing graph autoencoding performance across various scenarios.
Detailed accounts of the experimental setup, including the usage of Nvidia GPUs and the iterative optimization process, enhance reproducibility and clarity regarding the methodological rigor.
A deeper exploration of potential biases in the datasets or evaluation methods may warrant further scrutiny to ensure the validity of the conclusions drawn.

**Methods And Evaluation Criteria:**

The methods employed in the manuscript, including the structure of PieClam and the evaluation criteria for performance assessment, are appropriate for the problem domain. The authors effectively leveraged benchmark datasets for rigorous testing and comparison, enhancing the robustness of their findings. Specific elements to note include:
The incorporation of both inclusive and exclusive communities broadens the model's applicability to various graph types, making it adaptable for real-world scenarios.
The use of the Lorentz inner product for decoding, allowing a flexible representation of nodes in the affiliation space, is a strong methodological choice that lends depth to the investigation.
The section on experimental design and analyses confirms that the evaluation strategies align well with the objectives of the study, providing a solid foundation for claiming model effectiveness.

**Other Comments Or Suggestions:**

A careful proofread of the text could address minor typographical issues observed in sections discussing model evaluations.
The authors could consider including additional visualizations to further illustrate the model's capabilities and performance comparisons.

**Other Strengths And Weaknesses:**

Strengths of the paper include:
The originality of combining inclusive and exclusive communities to enhance node representation.
The clear presentation and methodological framework that makes the findings accessible and reproducible.
Weaknesses may involve:
The potential narrow focus on node affiliations without sufficiently addressing node features in edge conditions, which may limit applicability across diverse graph types.
A more elaborative discussion on model limitations would be beneficial.

**Questions For Authors:**

How do you envision incorporating node features into the edge probability calculations in future iterations of PieClam?
Could you elaborate on the potential applications of the log cut distance metric for sparse graphs, and your plans for addressing current limitations?
What specific challenges did you encounter during the implementation of the prior distribution learning, and how did you overcome them?

**Relation To Broader Scientific Literature:**

The contributions made by PieClam are significant within the landscape of graph representation learning. The use of overlapping community structures represents a notable advancement from predecessor models like BigClam and continue to resonate with core themes in community detection literature.
The model's introduction aligns with recent efforts to refine graph generative models and embed node representation learning in a probabilistic framework. Notably, the discussion of related works supports the authors' claim while presenting a complete narrative on how PieClam fits within the broader scientific discourse.

**Theoretical Claims:**

The theoretical claims, particularly regarding the model's universality and the correctness of the proofs, are adequately presented. Notable points include:
The completeness of the proofs for Theorems 3.7 and 3.8 highlights the rigorous mathematical grounding of the claims surrounding PieClam’s ability to approximate any graph utilizing a fixed parameter budget.
However, further discussion surrounding any assumptions made during the proof processes would strengthen the rigor of this section and provide additional assurance of the theoretical underpinnings.
Engagement with these theoretical proofs can help peer reviewers assess the significance and flexibility of the proposed method against existing literature.

---

> ### Author Rebuttal · Authors · 2025-03-30
>
> We thank that reviewer for the positive and in depth evaluation of our paper.
>
> >**Claims And Evidence:**
> >>...further clarity on the implications of using the log cut distance in sparse graphs would enhance the overall argument.
>
> *Response.*  Thank you for this suggestion.  We already shortly discussed this shortcoming in the Conclusion (Page 8):
>
> "Another limitation of our analysis is that the log cut distance is mainly appropriate for dense graphs. Future work will extend this metric to sparse graphs..."
>
> but we agree that this discussion may be too short. **If accepted, we will extend the discussion about sparsity, stating that the analysis is mostly appropriate for dense graphs, but the method is empirically appropriate also for sparse graphs.**
>
> Regarding applicability of the method to real-world sparse graphs, the method was tested against sparse graphs in anomaly detection and link prediction benchmark, where we got competitive performance, demonstrating that the method performs well empirically also for sparse graphs.
>
> >**Theoretical Claims:**
> >> However, further discussion surrounding any assumptions made during the proof processes would strengthen the rigor of this section...
>
> *Response.*  As a principle in mathematical writing, all assumptions are clearly and explicitly stated in the body of the theorem. No additional assumptions are allowed to be taken during the proof, as this would undermine the correctness of the theorem. We believe to have complied with this basic principle. However, if the reviewer found a hidden assumption in the body of the proof, please let us know where specifically this is made so we can fix it.
>
> >**Essential References Not Discussed:**
> >> The manuscript could benefit from including related works that ... explored universal properties of graph representations. .. Works exploring the strengths and weaknesses of using similar graph distance measures in sparse networks should also be cited.
>
> *Response.* We will add paragraphs to the  Extended Related Work appendix about this. We will cite papers about universality of GNNs as functions from graphs to vectors (graph classification or regression). For example:
>
> S. Chen, S. Lim, F. Memoli, Z. Wan, and Y. Wang. Weisfeiler-Lehman meets Gromov-Wasserstein. ICML. 2022.
>
> S. Chen, S. Lim, F. Memoli, Z. Wan, and Y. Wang. The Weisfeiler- Lehman distance: Reinterpretation and connection with GNNs.  Proceedings of 2nd Annual Workshop on Topology, Algebra, and Geometry in Machine Learning (TAG ML). 2023.
>
> J. B¨oker, R. Levie, N. Huang, S. Villar, and C. Morris. Fine-grained
> expressivity of graph neural networks. NeurIPS. 2023.
>
> L. Rauchwerger, S. Jegelka, R. Levie. Generalization, Expressivity, and Universality of Graph Neural Networks on Attributed Graphs. 2025.
>
> We will also add a short discussion about sparse graph similarity measures that can potentially be used to define a version of log-cut distance appropriate for sparse graphs.  For example:
>
> **Stretched Graphons:** C. Borgs, J. T. Chayes, H. Cohn, and N. Holden. Sparse exchangeable graphs and their limits via graphon processes. Journal of Machine Learning Research. 2018.
>
> **$L^p$ graphons:** C. Borgs, J. Chayes, H. Cohn, and Y. Zhao. A theory of sparse graph convergence i: Limits, sparse random graph models, and power law distributions. Transactions of the American Mathematical Society. 2019.
>
> **Graphings:** L. Lovász. Large networks and graph limits, volume 60. American Mathematical Soc., 2012.
>
> **Graphops:** Backhausz and B. Szegedy. Action convergence of operators and graphs. Canadian Journal of Mathematics, 74(1):72–121, 2022.
>
> >**Weaknesses:**
> >> The potential narrow focus on node affiliations without sufficiently addressing node features in edge conditions, which may limit applicability across diverse graph types...
>
> *Response.* First, we already addressed this issue in the Conclusion:
>
> *"One limitation of PieClam is that, for attributed graphs, it only models the node features through the prior in the community affiliation space, but not via the conditional probabilities of the edges (given the community affiliations). Future work will deal with extending PieClam to also include the node (or edge) features in the edge conditional probabilities."*
>
> Still, we often observe in experiments that not using the features can still lead to competitive performance with methods that do use the features. We wrote in section D3 of the appendix on page 25:
>
> *"Note that, since the prior is not used in classification, there is no added benefit from utilizing node features when training the prior, as
> proposed in 2.4. Therefore, the link prediction algorithm relies solely on the graph’s topological structure and the prior functions only as
> regularization."*
>
> **Nevertheless, we will extend this discussion in the camera-ready version of the paper if accepted.**

---

### Official Review · Reviewer_gDLV · 2025-03-17

**Overall Recommendation:** 4

**Summary:**

The paper introduces PieClam, a universal graph autoencoder that extends traditional community affiliation models by incorporating both inclusive and exclusive communities. The method uses a novel decoder based on the Lorentz inner product to overcome the triangle inequality limitations of previous models, thereby accurately representing graphs with structures like bipartite components;Also, the paper proposes a generative extension through a learned prior ) . The authors perform experiments on tasks such as anomaly detection and link prediction and show competitive performance against state-of-the-art baselines.

**Claims And Evidence:**

The authors claim that PieClam is a universal autoencoder that can approximate any graph (with a fixed parameter budget per node) and that it outperforms or competes with existing methods on several benchmarks. The paper supports these claims with both theoretical results (specifically , this can be shown in Theorems 3.7 and 3.8) and extensive experimental evaluations on synthetic datasets and real-world tasks. While the theoretical proofs appear sound under the presented assumptions, the evidence on sparse graphs is a bit less comprehensive. Further empirical validation in more diverse settings would improve the support for universality claims.

**Essential References Not Discussed:**

A couple of references that are relevant:
1. Chanpuriya et al. 2020. Node Embeddings and Exact Low-Rank Representations of Complex Networks
2. Jia et al. 2019.,CommunityGAN: Community Detection with Generative Adversarial Nets

**Experimental Designs Or Analyses:**

THe experimental design is thorough, that includes both synthetic and real-world datasets. The comparisons with several baselines in anomaly detection and link prediction are informative. Nonetheless, it would be useful to see more detailed analyses on the model’s performance with respect to computational efficiency and robustness to hyperparameter choices, especially since the training involves simultaneous optimization of node embeddings and a prior.

**Methods And Evaluation Criteria:**

I believe that the proposed method is well motivated and builds on prior models by extending the affiliation space to include exclusive communities. The use of the Lorentz inner product for decoding is a novel move that helps address challenges in previosu approaches. The evaluation (both synthetic experiments, anomaly detection, and link prediction) uses appropriate metrics and baselines. A minor comment is that, additional discussion on scalability (especially for large or sparse graphs) and hyperparameter sensitivity would be valuable.

**Other Comments Or Suggestions:**

Check my previous section.

**Other Strengths And Weaknesses:**

Strengths:
– Novel use of community affiliation models to include exclusive communities
– Theoretical guarantees of universality add significant depth to the contribution.
– Competitive empirical performant on multiple benchmarks

Weaknesses:
– The analysis appears more suited to dense graphs; applicability to very large, sparse graphs is less clear.
– The training procedure (involving simultaneous optimization of node embeddings and the generative prior) might be computationally demanding.
– Additional experiments on hyperparameter sensitivity and scalability would further strengthen the paper.

**Questions For Authors:**

Check my previous section.

**Relation To Broader Scientific Literature:**

The work is very relevant to the scientific literature.

**Theoretical Claims:**

The paper provides rigorous theoretical proofs of universality using the log cut distance framework. I checked the proofs for the main theoretical results (Theorems 3.7 and 3.8), and while they are largely convincing, some assumptions (e.g., regarding graph density) might limit their direct applicability to real-world sparse graphs. Clarifications on these limitations would improve the overall presentation.

---

> ### Author Rebuttal · Authors · 2025-03-30
>
> We thank the reviewer for the positive and in depth evaluation of our paper.
>
> >**Claims And Evidences:**
> >> ...the evidence on sparse graphs is a bit less comprehensive. Further empirical validation in more diverse settings would improve the support for universality claims.
>
> *Response.* The experiments in the current version of the paper (anomaly detection and link prediction) are already on sparse graphs. For example, the Texas graph has 183 nodes and 279 edges.
>
> We ran additional experiments on link prediction on the OGB-ddi dataset. We will add more experiments in the final version.
>
> | Model                  | Hits@20 test         | AUC
> |------------------------|----------------------|-------------
> | **IeClam**             | **90.72 ± 2.35**     | 99.89 ± 0.00
> | NCN                    | 76.52 ± 10.47        | 99.97 ± 0.00
> | NCNC                   | 70.23 ± 12.11        | 99.97 ± 0.01
> | GraphSAGE              | 49.84 ± 15.56        | 99.96 ± 0.00
> | GCN                    | 49.90 ± 7.23         | 99.86 ± 0.03
> | SEAL                   | 25.25 ± 3.90         | 97.97 ± 0.19
> | PEG                    | 30.28 ± 4.92         | 99.45 ± 0.04
> | BUDDY                  | 29.60 ± 4.75         | 99.81 ± 0.02
> | Node2Vec               | 34.69 ± 2.90         | 99.78 ± 0.04
> | MF                     | 23.50 ± 5.35         | 99.46 ± 0.10
> | Neo-GNN                | 20.95 ± 6.03         | 98.06 ± 2.00
> | GAT                    | 31.88 ± 8.83         | 99.63 ± 0.21
> | CN                     | 17.73                | 95.20
> | AA                     | 18.61                | 95.43
> | RA                     | 6.23                 | 96.51
> | Shortest Path          | 0                    | 59.07
>
> >**Methods And Evaluation Criteria:**
> >>...additional discussion on scalability (especially for large or sparse graphs) and hyperparameter sensitivity would be valuable.
>
> *Response.* Regarding scalability, we will add to the camera ready version of the paper a discussion that shows that both the memory and time complexity of all Clam models are linear in the number of edges.
>
> We already wrote about the computational complexity of BigClam on Page 3, Column 1, Line 144:
>
> *"In order to implement the above iteration with $O(|E|)$ operations at each step, instead of $O(N^2)$, the loss can be rearranged as..."*
>
> Moreover we wrote about BigClam, Page 3, Column 1, Line 158:
>
> *"We observe that the optimization process is a message passing scheme"*
>
> We agree that we have not sufficiently clarified that all other Clam models are trained with a message passing scheme, and share their complexity with BigClam: linear in the number of edges. We only wrote on Page 4, Column1, Line 208:
>
> *"This loss can be efficiently implemented on sparse graph by the formulation..."*
>
> **We will clarify in the camera ready version that all Clam models have linear complexity with respect to the number of edges.**
>
> Regarding hyperparameter sensitivity, in Appendix D6 we do an ablation study where we show the impact of certain parameters on the performance of the model. We can extend this appendix to study sensitivity to more hyperparameters.
>
> >**Theoretical Claims:**
> >>...some assumptions (e.g., regarding graph density) might limit their direct applicability to real-world sparse graphs.
>
> *Response.* We already shortly discussed this in the Conclusion (Page 8):
>
> "Another limitation of our analysis is that the log cut distance is mainly appropriate for dense graphs. Future work will extend this metric to sparse graphs..."
>
> but we agree that this discussion may be too short. **If accepted, we can extend the discussion about sparsity, mainly stating that the analysis is mostly appropriate for dense graphs (but the method is appropriate also for sparse graphs).**
>
> Regarding applicability of the method to real-world graphs, the method was tested against sparse graphs in anomaly detection and link prediction benchmark, where we got competitive performance, demonstrating that the method performs well empirically also for sparse graphs.
>
>
> >**Experimental Designs Or Analyses:**
> >>Nonetheless, it would be useful to see more detailed analyses on the model’s performance with respect to computational efficiency... especially since the training involves simultaneous optimization of node embeddings and a prior.
>
>
> *Response.* Note that the prior is a node-wise computation, so it is dominated by the rest of the log likelihood terms in optimization.
>
> We wrote on Page 4, Column 2, Like 216:
>
> *"Observe that the PieClam loss is similar to the IeClam loss, only with the addition of the prior acting as a per node regularization term"*
>
> We will extend the text in the camera-ready version and write that, as a reuslt, adding the prior does not asymptotically increase the complexity of the model.
>
> >**Essential References Not Discussed:**
>
> We will properly refer to the suggested papers in the camera-ready version of the paper if accepted.
>
> >**Weaknesses:**
>
> *Response.* See our responses above.

---

### Official Review · Reviewer_3NUL · 2025-03-17

**Overall Recommendation:** 3

**Summary:**

The submitted manuscript introduces PieClam, a graph autoencoder that learns node embeddings by maximizing the log-likelihood of the decoded graph. It extends the well-known BigClam method in two ways. First, PieClam incorporates a learned prior on the node distribution in the embedding space, shifting from a simple pairwise interaction approach to a full-fledged graph generative model. Second, while it retains BigClam’s focus on sets of nodes with high connectivity (inclusive communities), it also introduces the notion of exclusive communities, i.e., groups of nodes that exhibit strong disconnection. This dual capacity to identify both inclusive and exclusive communities is enabled by a new graph similarity measure called the log cut distance, through which the authors demonstrate that PieClam is a universal autoencoder capable of approximating any graph distribution within a uniform bound. Empirical results are provided in tasks such as graph anomaly detection and link prediction.

**Claims And Evidence:**

The claims made in the submission are supported by clear and convincing evidence.

**Essential References Not Discussed:**

The reference list is sufficient.

**Experimental Designs Or Analyses:**

I carefully reviewed the experimental setup and the corresponding analyses, which appear reasonable.

**Methods And Evaluation Criteria:**

Yes. The paper’s focus on discovering both inclusive and exclusive communities naturally lends itself to tasks that test how well a model captures nuanced graph structure. Evaluating on anomaly detection and link prediction is appropriate because these tasks directly assess whether the learned embeddings and generative assumptions effectively capture both the presence and absence exclusive of edges.

**Other Comments Or Suggestions:**

NA

**Other Strengths And Weaknesses:**

Strengths

1. The paper demonstrates notable originality by extending BigClam into a generative framework and enables a richer representation of graph structure compared to BigClam.
2. The theoretical analysis is thorough and convincingly supports the approach, underscoring its mathematical soundness.
3. The manuscript is well written, clearly organized, and technically solid, making it accessible to both experts and newcomers in the field.

Weaknesses

1. A key concern lies in the equivariance of the encoder architecture. While it is straightforward to design universal autoencoders when equivariance is not enforced, the paper does not clarify how its approach balances equivariance against universality.
2. The experimental evaluation would benefit from additional benchmarks, particularly for link prediction on OGB datasets, to more rigorously establish the method’s effectiveness and generalizability.
3. The computational complexity of the proposed approach remains unspecified, leaving questions about its scalability and feasibility for large datasets.
4. The paper does not fully explain how the model generalizes to unseen data.

**Questions For Authors:**

Please see respond to the aforementioned weaknesses.

**Relation To Broader Scientific Literature:**

PieClam extends ideas from BigClam’s community affiliation framework by adding a learned prior to make it a generative model, drawing on concepts used in SBMs that define latent distributions for edge formation. In embracing both inclusive and exclusive communities, it generalizes beyond models that preserve the triangle inequality and can be applied to bipartite graphs. PieClam’s log cut distance builds upon well-known cut-based measures in graph theory and helps establish it as a universal autoencoder.

**Theoretical Claims:**

The theoretical claims appear to be correct, although I have not personally verified the proofs in detail.

---

> ### Author Rebuttal · Authors · 2025-03-30
>
> We thank the reviewer for the positive and in depth evaluation of our paper.
>
> >**Weaknesses:**
> >>1. A key concern lies in the equivariance of the encoder architecture...
>
> *Response.* Thank you for this comment. **PieClam (and all other Clam models) are in fact equivariant to node re-indexing.** This is due to the fact that  maximizing the log likelihood via gradient descent in Clam models can be formulated as a message passing algorithm. We wrote this on Page 3, Column 1, Line 158 (about BigClam):
>
> *"We observe that the optimization process is a message passing scheme"*
>
> But we indeed did not link this to equivariance, and did not repeat this claim about all other Clam models. **If accepted, we will add an explanation about the equivariance of Clam methods to node re-indexing.**
>
> >>2. The experimental evaluation would benefit from additional benchmarks...
>
> *Response.* We ran additional experiments on link prediction on the OGB-ddi dataset. We will add more experiments in the final version.
>
> | Model                  | Hits@20 test         | AUC
> |------------------------|----------------------|-------------
> | **IeClam**             | **90.72 ± 2.35**     | 99.89 ± 0.00
> | NCN                    | 76.52 ± 10.47        | 99.97 ± 0.00
> | NCNC                   | 70.23 ± 12.11        | 99.97 ± 0.01
> | GraphSAGE              | 49.84 ± 15.56        | 99.96 ± 0.00
> | GCN                    | 49.90 ± 7.23         | 99.86 ± 0.03
> | SEAL                   | 25.25 ± 3.90         | 97.97 ± 0.19
> | PEG                    | 30.28 ± 4.92         | 99.45 ± 0.04
> | BUDDY                  | 29.60 ± 4.75         | 99.81 ± 0.02
> | Node2Vec               | 34.69 ± 2.90         | 99.78 ± 0.04
> | MF                     | 23.50 ± 5.35         | 99.46 ± 0.10
> | Neo-GNN                | 20.95 ± 6.03         | 98.06 ± 2.00
> | GAT                    | 31.88 ± 8.83         | 99.63 ± 0.21
> | CN                     | 17.73                | 95.20
> | AA                     | 18.61                | 95.43
> | RA                     | 6.23                 | 96.51
> | Shortest Path          | 0                    | 59.07
>
> >>3. The computational complexity of the proposed approach remains unspecified, leaving questions about its scalability and feasibility for large datasets.
>
> *Response.* We wrote about the computational complexity of BigClam on Page 3, Column 1, Line 144:
>
> *"In order to implement the above iteration with $O(|E|)$ operations at each step, instead of $O(N^2)$, the loss can be rearranged as..."*
>
> As written above, basically, Clam models are trained with message passing algorithms, so they have efficient computational complexity: linear in the number of edges.
>
> BigClam. On Page 4, Column 1, Line 208, wewrote:
>
> *"This loss can be efficiently implemented on sparse graph by the formulation..."*
>
> We agree that we have not clarified enough that all other Clam models, including PieClam, share their computational and memory complexity with
>
> **We will clarify in the camera ready version (if accepted) that all Clam models have linear complexity with respect to the number of edges**
>
> >>4. The paper does not fully explain how the model generalizes to unseen data.
>
> *Response.* Studying how and why models generalize to unseen data is one of the key and most fundamental (mostly open) questions in learning theory, and specifically statistical learning. In this paper we do not provide generalization analysis, e.g., in a PAC learnability setting, VC dimension, Rademacher complexity, etc. We leave such analysis to future works, and note that most papers that first proposes a deep learning architecture do not provide a generalization analysis.
>
> If your question is meant to be more practical, i.e., how the training and test data are defined, and how testing is defined, this depends on the problem.
>
> 1. In anomaly detection, the setting is explained in Section 4.3:
>
> *"In unsupervised node anomaly detection, one is given a graph with node features, where some of the nodes are unknown anomalies. The goal is to detect these anomalous nodes, without supervising on any example of normal or anomalous nodes, using only the structure of the graph and the node features"*
>
> Namely, here generalization is better interpreted as a form of "transfer learning". One fits the Clam model to the whole graph data (solving the task of reconstruction/autoencoder), and then uses the trained community affiliation features to solve a different task: evaluating the likelihood of nodes according to the learned probabilistic graph model.
>
> 2.  In link prediction, as explained in Section 4.4:
>
> *"In supervised link prediction, one is given a graph where for some of the dyads it is unknown if they are edges or not. The goal is to predict the connectivity of the omitted dyads."*
>
> So, here the training set for he Clam model are the known dyads (the given known edges and non-edges) and the test set are the unknown dyads, where the task is to decide if each unknown dyad is an edge or a non-edge.

---

### Decision · Program_Chairs · 2025-05-01

**Decision:**

Accept (poster)

**Comment:**

The manuscript introduces PieClam, a graph autoencoder that extends BigClam by incorporating a learned prior over node embeddings, making it a generative model. It also introduces exclusive communities, representing disconnected node groups alongside traditional inclusive ones. Using a new similarity metric, log cut distance, the authors prove PieClam is a universal autoencoder capable of approximating any graph. Experiments on anomaly detection and link prediction demonstrate its strong performance.

The paper is of clear good quality and a solid contribution to ICML.